# MLPInit: Embarrassingly Simple GNN Training Acceleration with MLP Initialization

**Xiaotian Han**[1]*  **Tong Zhao**[2]  **Yozen Liu**[2]  **Xia Hu**[3]  **Neil Shah**[2]
[1]Texas A&M University    [2]Snap Inc.    [3]Rice University
han@tamu.edu   {tzhao,yliu2,nshah}@snap.com   xia.hu@rice.edu

## Abstract

Training graph neural networks (GNNs) on large graphs is complex and extremely time consuming. This is attributed to overheads caused by sparse matrix multiplication, which are sidestepped when training multi-layer perceptrons (MLPs) with only node features. MLPs, by ignoring graph context, are simple and faster for graph data, however they usually sacrifice prediction accuracy, limiting their applications for graph data. We observe that for most message passing-based GNNs, we can trivially derive an analog MLP (we call this a PeerMLP) with an equivalent weight space, by setting the trainable parameters with the same shapes, making us curious about *how do GNNs using weights from a fully trained **PeerMLP** perform?* Surprisingly, we find that GNNs initialized with such weights significantly outperform their PeerMLPs, motivating us to use PeerMLP training as a precursor, initialization step to GNN training. To this end, we propose an embarrassingly simple, yet hugely effective initialization method for GNN training acceleration, called MLPInit. Our extensive experiments on multiple large-scale graph datasets with diverse GNN architectures validate that MLPInit can accelerate the training of GNNs (up to 33× speedup on `OGB-products`) and often improve prediction performance (e.g., up to 7.97% improvement for GraphSAGE across 7 datasets for node classification, and up to 17.81% improvement across 4 datasets for link prediction on metric Hits@10). The code is available at https://github.com/snap-research/MLPInit-for-GNNs.

## 1 Introduction

Graph Neural Networks (GNNs) (Zhang et al., 2018; Zhou et al., 2020; Wu et al., 2020) have attracted considerable attention from both academic and industrial researchers and have shown promising results on various practical tasks, e.g., recommendation (Fan et al., 2019; Sankar et al., 2021; Ying et al., 2018; Tang et al., 2022), knowledge graph analysis (Arora, 2020; Park et al., 2019; Wang et al., 2021), forecasting (Tang et al., 2020; Zhao et al., 2021; Jiang & Luo, 2022) and chemistry analysis (Li et al., 2018b; You et al., 2018; De Cao & Kipf, 2018; Liu et al., 2022). However, training GNN on large-scale graphs is extremely time-consuming and costly in practice, thus spurring considerable work dedicated to scaling up the training of GNNs, even necessitating new massive graph learning libraries (Zhang et al., 2020; Ferludin et al., 2022) for large-scale graphs.

Recently, several approaches for more efficient GNNs training have been proposed, including novel architecture design (Wu et al., 2019; You et al., 2020d; Li et al., 2021), data reuse and partitioning paradigms (Wan et al., 2022; Fey et al., 2021; Yu et al., 2022) and graph sparsification (Cai et al., 2020; Jin et al., 2021b). However, these kinds of methods often sacrifice prediction accuracy and increase modeling complexity, while sometimes meriting significant additional engineering efforts.

MLPs are used to accelerate GNNs (Zhang et al., 2021b; Frasca et al., 2020; Hu et al., 2021) by decoupling GNNs to node features learning and graph structure learning. Our work also leverages MLPs but adopts a distinct perspective. Notably, we observe that the weight space of MLPs and GNNs can be identical, which enables us to transfer weights between MLP and GNN models. Having the fact that MLPs train faster than GNNs, this observation inspired us to raise the question:

---
*This work was done while the first author was an intern at Snap Inc.

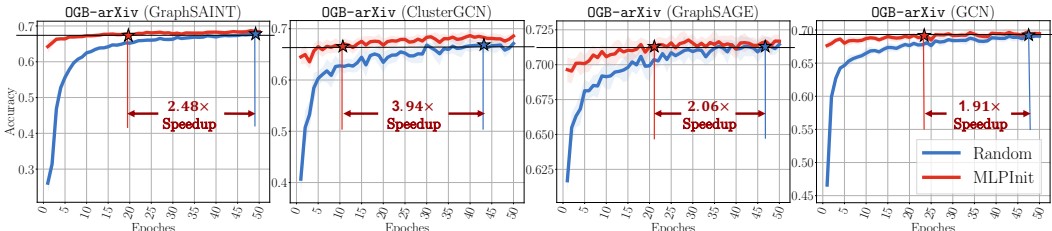

Figure 1: The training speed comparison of the GNNs with Random initialization and MLPInit. ★ indicates the best performance that GNNs with random initialization can achieve. ★ indicates the comparable performance of the GNN with MLPInit. Speedup indicates the training time reduced by our proposed MLPInit compared to random initialization. This experimental result shows that MLPInit is able to accelerate the training of GNNs significantly.

### *Can we train GNNs more efficiently by leveraging the weights of converged MLPs?*

To answer this question, we first pioneer a thorough investigation to reveal the relationship between the MLPs and GNNs in terms of trainable weight space. For ease of presentation, we define the PeerMLP of a GNN[1] so that GNN and its PeerMLP share the same weights [2]. We find that interestingly, *GNNs can be optimized by training the weights of their PeerMLP*. Based on this observation, we adopt weights of converged PeerMLP as the weights of corresponding GNNs and find that these GNNs perform even better than converged PeerMLP on node classification tasks (results in Table 2).

Motivated by this, we propose an embarrassingly simple, yet remarkably effective method to accelerate GNNs training by initializing GNN with the weights of its converged PeerMLP. Specifically, to train a target GNN, we first train its PeerMLP and then initialize the GNN with the optimal weights of converged PeerMLP. We present the experimental results in Figure 1 to show the training speed comparison of GNNs with random initialization and with MLPInit. In Figure 1, Speedup shows the training time reduced by our proposed MLPInit compared to random initialized GNN, while achieving the same test performance. This experimental result shows that MLPInit is able the accelerate the training of GNNs significantly: for example, we speed up the training of GraphSAGE, GraphSAINT, ClusterGCN, GCN by $2.48\times$, $3.94\times$, $2.06\times$, $1.91\times$ on OGB-arXiv dataset, indicating the superiority of our method in GNNs training acceleration. Moreover, we speed up GraphSAGE training more than $14\times$ on OGB-products. We highlight **our contributions** as follows:

- We pioneer a thorough investigation to reveal the relationship between MLPs and GNNs in terms of the trainable weight space through the following observations: *(i)* GNNs and MLPs have the same weight space. *(ii)* GNNs can be optimized by training the weights of their PeerMLPs. *(iii)* GNN with weights from its converged PeerMLP surprisingly performs better than the performance of its converged PeerMLP on node classification tasks.

- Based on the above observations, we proposed an embarrassingly simple yet surprisingly effective initialization method to accelerate the GNNs training. Our method, called MLPInit, initializes the weights of GNNs with the weight of their converged PeerMLP. After initialization, we observe that GNN training takes less than half epochs to converge than those with random initialization. Thus, MLPInit is able to accelerate the training of GNNs since training MLPs is cheaper and faster than training GNNs.

- Comprehensive experimental results on multiple large-scale graphs with diverse GNNs validate that MLPInit is able to accelerate the training of GNNs (up to $33\times$ speedup on OGB-products) while often improving the model performance [3] (e.g., $7.97\%$ improvement for node classification on GraphSAGE and $17.81\%$ improvement for link prediction on Hits@10).

- MLPInit is extremely easy to implement and has virtually negligible computational overhead compared to the conventional GNN training schemes. In addition, it is orthogonal to other GNN acceleration methods, such as weight quantization and graph coarsening, further increasing headroom for GNN training acceleration in practice.

---

[1]The formal definition of PeerMLP is in Section 3.

[2]By *share the same weight*, we mean that the trainable weights of GNN and its PeerMLP are the same in terms of size, dimension, and values.

[3]By *performance*, we refer to the model prediction quality metric of the downstream task on the corresponding test data throughout the discussion.

## 2 PRELIMINARIES

**Notations**. We denote an attributed graph $\mathcal{G} = (\mathbf{X}, \mathbf{A})$, where $\mathbf{X} = \{\mathbf{x}_1, \mathbf{x}_2, \cdots, \mathbf{x}_N\} \in \mathbb{R}^{N \times D}$ is the node feature matrix and $\mathbf{A} = \{0, 1\}^{N \times N}$ is the binary adjacency matrix. $N$ is the number of nodes, and $D$ is the dimension of node feature. For the node classification task, we denote the prediction targets by $\mathbf{Y} \in \{0, 1, \ldots, C-1\}^N$, where $C$ is the number of classes. We denote a GNN model as $f_{gnn}(\mathbf{X}, \mathbf{A}; w_{gnn})$, and an MLP as $f_{mlp}(\mathbf{X}; w_{mlp})$, where $w_{gnn}$ and $w_{mlp}$ denote the trainable weights in the GNN and MLP, respectively. Moreover, $w_{gnn}^*$ and $w_{mlp}^*$ denote the fixed weights of optimal (or converged) GNN and MLP, respectively.

**Graph Neural Networks**. Although various forms of graph neural networks (GNNs) exist, our work refers to the conventional message passing flavor (Gilmer et al., 2017). These models work by learning a node's representation by aggregating information from the node's neighbors recursively. One simple yet widely popular GNN instantiation is the graph convolutional network (GCN), whose multi-layer form can be written concisely: the representation vectors of all nodes at the $l$-th layer are $\mathbf{H}^l = \sigma(\mathbf{A}\mathbf{H}^{l-1}\mathbf{W}^l)$, where $\sigma(\cdot)$ denotes activation function, $\mathbf{W}^l$ is the trainable weights of the $l$-th layer, and $\mathbf{H}^{l-1}$ is the node representations output by the previous layer. Denoting the output of the last layer of GNN by $\mathbf{H}$, for a node classification task, the prediction of node label is $\hat{\mathbf{Y}} = \text{softmax}(\mathbf{H})$. For a link prediction task, one can predict the edge probabilities with any suitable decoder, e.g., commonly used inner-product decoder as $\hat{\mathbf{A}} = \text{sigmoid}(\mathbf{H} \cdot \mathbf{H}^T)$ (Kipf & Welling, 2016b).

## 3 MOTIVATING ANALYSES

In this section, we reveal that MLPs and GNNs share the same weight space, which facilitates the transferability of weights between the two architectures. Through this section, we use GCN (Kipf & Welling, 2016a) as a prototypical example for GNNs for notational simplicity, but we note that our discussion is generalizable to other message-passing GNN architectures.

**Motivation 1: GNNs share the same weight space with MLPs**. To show the weight space of GNNs and MLPs, we present the mathematical expression of one layer of MLP and GCN (Kipf & Welling, 2016a) as follows:

$$\text{GNN:} \quad \mathbf{H}^l = \sigma(\mathbf{A}\mathbf{H}^{l-1}\mathbf{W}_{gnn}^l), \qquad \text{MLP:} \quad \mathbf{H}^l = \sigma(\mathbf{H}^{l-1}\mathbf{W}_{mlp}^l), \qquad (1)$$

where $\mathbf{W}_{gnn}^l$ and $\mathbf{W}_{mlp}^l$ are the trainable weights of $l$-th layer of MLP and GCN, respectively. If we set the hidden layer dimensions of GNNs and MLPs to be the same, then $\mathbf{W}_{mlp}^l$ and $\mathbf{W}_{gnn}^l$ will naturally have the same size. Thus, although the GNN and MLP are different models, their weight spaces can be identical. Moreover, for any GNN model, we can trivially derive a corresponding MLP whose weight space can be made identical. For brevity, and when the context of a GNN model is made clear, we can write such an MLP which shares the same weight space as a PeerMLP, i.e., their trainable weights can be transferred to each other.

**Motivation 2: MLPs train faster than GNNs**. GNNs train slower than MLPs, owing to their non-trivial relational data dependency. We empirically validate that training MLPs is much faster than training GNNs in Table 1. Specifically, this is because MLPs do not involve sparse matrix multiplication for neighbor aggregation. A GNN layer (here we consider a simple GCN layer, as defined in Equation (1)) can be broken down into two operations: feature transformation ($\mathbf{Z} = \mathbf{H}^{l-1}\mathbf{W}^l$) and neighbor aggregation ($\mathbf{H}^l = \mathbf{A}\mathbf{Z}$) (Ma et al., 2021). The neighbor aggregation and feature transformation are typically sparse and dense matrix multiplications, respectively. Table 1 shows the time usage for these different operations on several real-world graphs. As expected, neighbor aggregation in GNNs consumes the large majority of computation time. For example, on the Yelp dataset, the neighbor aggregation operation induces a 3199× time overhead.

Given that the weights of GNNs and their PeerMLP can be transferred to each other, but the PeerMLP can be trained much faster, we raise the following questions:

1. ***What will happen if we directly adopt the weights of a converged PeerMLP to GNN?***
2. ***To what extent can PeerMLP speed up GNN training and improve GNN performance?***

In this paper, we try to answer these questions with a comprehensive empirical analysis.

Table 1: Comparison of the running time of forward and backward for different operations (i.e., feature transformation and neighbor aggregation) in GNNs. The time unit is milliseconds (*ms*).

| Operation | OGB-arXiv | | | Flickr | | | Yelp | | |
|---|---|---|---|---|---|---|---|---|---|
| #Nodes
#Edges | 169343
1166243 | | | 89250
899756 | | | 716847
13954819 | | |
| | Forward | Backward | Total | Forward | Backward | Total | Forward | Backward | Total |
| $Z = XW$ | 0.32 | 1.09 | 1.42 | 0.28 | 0.97 | 1.26 | 1.58 | 4.41 | 5.99 |
| $H = AZ$ | 1.09 | 1028.08 | 1029.17 | 1.01 | 836.95 | 837.97 | 9.74 | 19157.17 | 19166.90 |
| | | | 724× | | | 665× | | | 3199× |

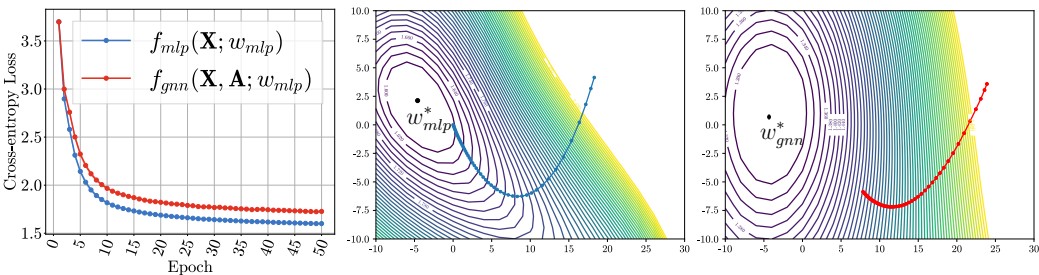

Figure 2: The relation of GNN and MLP during the training of PeerMLP. **Left**: Cross-Entropy loss of $f_{gnn}(\mathbf{X}, \mathbf{A}; w_{mlp})$ (GNN) and $f_{mlp}(\mathbf{X}; w_{mlp})$ (PeerMLP) on training set over training epochs of PeerMLP. In this experiment, GNN and PeerMLP share the same weight $w_{mlp}$, which are trained by the PeerMLP. **Middle**: training trajectory of PeerMLP on its own loss landscape. **Right**: training trajectory of GNN with weights from PeerMLP on GNN's loss landscape. The figures show that training loss of GNN with weights trained from MLP will decrease. The details are presented in Appendix D.2. We also present loss curves on validation/test sets and accuracy curves in Appendix A.4.

## 4 WHAT WILL HAPPEN IF WE DIRECTLY ADOPT THE WEIGHTS OF A CONVERGED PEERMLP TO GNN?

To answer this question, we conducted comprehensive preliminary experiments to investigate weight transferability between MLPs and GNNs. We made the following interesting and inspiring findings:

**Observation 1: The training loss of GNN will decrease by optimizing the weights of its PeerMLP**. We conducted a verification experiment to investigate the loss changes of the GNNs with the weights trained from its PeerMLP and present the results in Figure 2. In this experiment, we have two models, a GNN and its corresponding PeerMLP, who share the same weights $w_{mlp}$. That is, the PeerMLP is $f_{mlp}(\mathbf{X}; w_{mlp})$ and the GNN is $f_{gnn}(\mathbf{X}, \mathbf{A}; w_{mlp})$. We optimize the weights $w_{mlp}$ by training the PeerMLP, and the loss curve of $f_{mlp}(\mathbf{X}; w_{mlp})$ is the blue line in the left figure in Figure 2. We also compute the loss of GNN $f_{gnn}(\mathbf{X}, \mathbf{A}; w_{mlp})$ with the weights from PeerMLP. The loss curve of $f_{gnn}(\mathbf{X}, \mathbf{A}; w_{mlp})$ is shown in the red line. Figure 2 shows the surprising phenomenon that the training loss of GNN with weights trained from PeerMLP decreases consistently. Impressively, these weights ($w_{mlp}$) were derived without employing neighbor aggregation in training.

**Observation 2: Converged weights from PeerMLP provide a good GNN initialization.** As PeerMLP and GNN have the same weight spaces, a natural follow-up question is whether GNN can directly adopt the weights of the converged PeerMLP and perform well. We next aim to understand this question empirically. Specifically, we first trained a PeerMLP for a target GNN and obtained the optimal weights $w_{mlp}^*$. Next, we run inference on test data using a GNN with $w_{mlp}^*$ of PeerMLP, i.e., applying $f_{gnn}(\mathbf{X}, \mathbf{A}; w_{mlp}^*)$. Table 2 shows the results of $f_{mlp}(\mathbf{X}; w_{mlp}^*)$ and $f_{gnn}(\mathbf{X}, \mathbf{A}; w_{mlp}^*)$. We can observe that the

Table 2: The performance of GNNs and its PeerMLP with the weights of a converged PeerMLP on test data.

| | Methods | PeerMLP | GNN | Improv. | MLPInit |
|---|---|---|---|---|---|
| OGB-arXiv | GraphSAGE | 56.04±0.27 | 62.87±0.95 | ↑ 12.18% | 72.25±0.30 |
| | GraphSAINT | 53.88±0.41 | 63.26±0.71 | ↑ 17.41% | 68.80±0.20 |
| | ClusterGCN | 54.47±0.41 | 60.81±1.30 | ↑ 11.63% | 69.53±0.50 |
| | GCN | 56.31±0.21 | 56.28±0.89 | ↓ 0.04% | 70.35±0.34 |
| OGB-products | GraphSAGE | 63.43±0.14 | 74.32±1.04 | ↑ 17.16% | 80.04±0.62 |
| | GraphSAINT | 57.29±0.32 | 69.00±1.54 | ↑ 20.44% | 74.02±0.19 |
| | ClusterGCN | 59.53±0.46 | 71.74±0.70 | ↑ 20.51% | 78.48±0.64 |
| | GCN | 62.63±0.15 | 71.11±0.10 | ↑ 13.55% | 76.85±0.34 |

GNNs with the optimal weights of PeerMLP consistently outperform PeerMLP, indicating that the weights from converged PeerMLP can serve as good enough initialization of the weights of GNNs.

## 4.1 THE PROPOSED METHOD: MLPINIT

The above findings show that MLPs can help the training of GNNs. In this section, we formally present our method MLPInit, which is an embarrassingly simple, yet extremely effective approach to accelerating GNN training.

The basic idea of MLPInit is straightforward: we adopt the weights of a converged PeerMLP to initialize the GNN, subsequently, fine-tune the GNN. Specifically, for a target GNN ($f_{gnn}(\mathbf{X}, \mathbf{A}; w_{gnn})$), we first construct a PeerMLP ($f_{mlp}(\mathbf{X}, \mathbf{A}; w_{mlp})$), with matching target weights. Next, we optimize the weight of the PeerMLP model by training the PeerMLP solely with the node features $\mathbf{X}$ for $m$ epochs. Upon training the PeerMLP to convergence and obtaining the optimal weights ($w_{mlp}^*$), we initialize the GNN with $w_{mlp}^*$ and then fine-tune the GNN with $n$ epochs. We present PyTorch-style pseudo-code of MLPInit in node classification setting in Algorithm 1.

**Training Acceleration**. Since training of the PeerMLP is comparatively cheap, and the weights of the converged PeerMLP can provide a good initialization for the corresponding GNN, the end result is that we can significantly reduce the training time of the GNN. Assuming that the training of GNN from a random initialization needs $N$ epochs to converge, and $N >> n$, the total training time can be largely reduced given that MLP training time is negligible compared to GNN training time. The experimental results in Table 3 show that $N$ is generally much larger than $n$.

**Algorithm 1** PyTorch-style Pseudocode of MLPInit

```
# f_gnn: graph neural network model
# f_mlp: PeerMLP of f_gnn

# Train PeerMLP for N epochs
for X, Y in dataloader_mlp:
    P = f_mlp(X)
    loss = nn.CrossEntropyLoss(P, Y)
    loss.backward()
    optimizer_mlp.step()

# Initialize GNN with MLPInit
torch.save(f_mlp.state_dict(), "w_mlp.pt")
f_gnn.load_state_dict("w_mlp.pt")

# Train GNN for n epochs
for X, A, Y in dataloader_gnn:
    P = f_gnn(X, A)
    loss = nn.CrossEntropyLoss(P, Y)
    loss.backward()
    optimizer_gnn.step()
```

**Ease of Implementation**. MLPInit is extremely easy to implement as shown in Algorithm 1. First, we construct an MLP (PeerMLP), which has the same weights with the target GNN. Next, we use the node features $\mathbf{X}$ and node labels $\mathbf{Y}$ to train the PeerMLP to converge. Then, we adopt the weights of converged PeerMLP to the GNN, and fine-tune the GNN while additionally leveraging the adjacency $\mathbf{A}$. In addition, our method can also directly serve as the final, or deployed GNN model, in resource-constrained settings: assuming $n = 0$, we can simply train the PeerMLP and adopt $w_{mlp}^*$ directly. This reduces training cost further, while enabling us to serve a likely higher performance model in deployment or test settings, as Table 2 shows.

## 4.2 DISCUSSION

In this section, we discuss the relation between MLPInit and existing methods. Since we position MLPInit as an acceleration method involved MLP, we first compare it with MLP-based GNN acceleration methods, and we also compare it with GNN Pre-training methods.

**Comparison to MLP-based GNN Acceleration Methods**. Recently, several works aim to simplify GNN to MLP-based constructs during training or inference (Zhang et al., 2022; Wu et al., 2019; Frasca et al., 2020; Sun et al., 2021; Huang et al., 2020; Hu et al., 2021). Our method is proposed to accelerate the message passing based GNN for large-scale graphs. Thus, MLP-based GNN acceleration is a completely different line of work compared to ours since it removes the message passing in the GNNs and uses MLP to model graph structure instead. Thus, MLP-based GNN acceleration methods are out of the scope of the discussion in this work.

**Comparison to GNN Pre-training Methods**. Our proposed MLPInit are orthogonal to the GNN pre-training methods(You et al., 2020b; Zhu et al., 2020b; Veličković et al., 2018b; You et al., 2021; Qiu et al., 2020; Zhu et al., 2021; Hu et al., 2019). GNN pre-training typically leverages graph augmentation to pretrain weights of GNNs or obtain the node representation for downstream tasks. Compared with the pre-training methods, MLPInit has two main differences (or advantages) that significantly contribute to the speed up: (i) the training of PeerMLP does not involve using the graph structure data, while pre-training methods rely on it. (ii) Pre-training methods usually involve graph data augmentation (Qiu et al., 2020; Zhao et al., 2022a), which requires additional training time.

Table 3: Speed improvement when MLPInit achieves comparable performance with random initialized GNN. The number reported is the training epochs needed. (—) means our method can not reach comparable performance. The epoch used by Random/MLPInit is denoted as ⭐/⭐ in Figure 1. The detailed speedup computation method are presented in Appendix D.3.

| | Methods | Flickr | Yelp | Reddit | Reddit2 | A-products | OGB-arXiv | OGB-products | Avg. |
|---|---|---|---|---|---|---|---|---|---|
| **SAGE** | Random(⭐) | 45.6 | 44.7 | 36.0 | 48.0 | 48.9 | 46.7 | 43.0 | 44.7 |
| | MLPInit (⭐) | 39.9 | 20.3 | 7.3 | 7.7 | 40.8 | 22.7 | 2.9 | 20.22 |
| | Improv. | 1.14× | 2.20× | 4.93× | 6.23× | 1.20× | 2.06× | 14.83× | 2.21× |
| **SAINT** | Random | 31.0 | 35.8 | 40.6 | 28.3 | 50.0 | 48.3 | 44.9 | 40.51 |
| | MLPInit | 14.1 | 0.0 | 21.8 | 6.1 | 9.1 | 19.5 | 16.9 | 14.58 |
| | Improv. | 2.20× | — | 1.86× | 4.64× | 5.49× | 2.48× | 2.66× | 2.77× |
| **C-GCN** | Random | 15.7 | 40.3 | 46.2 | 47.0 | 37.4 | 42.9 | 42.8 | 38.9 |
| | MLPInit | 7.3 | 18.0 | 12.8 | 17.0 | 1.0 | 10.9 | 15.0 | 11.7 |
| | Improv. | 2.15× | 2.24× | 3.61× | 2.76× | 37.40× | 3.94× | 2.85× | 3.32× |
| **GCN** | Random | 46.4 | 44.5 | 42.4 | 2.4 | 47.7 | 46.7 | 43.8 | 45.35 |
| | MLPInit | 30.5 | 23.3 | 0.0 | 0.0 | 0.0 | 24.5 | 1.3 | 19.9 |
| | Improv. | 1.52× | 1.91× | — | — | — | 1.91× | 33.69× | 2.27× |

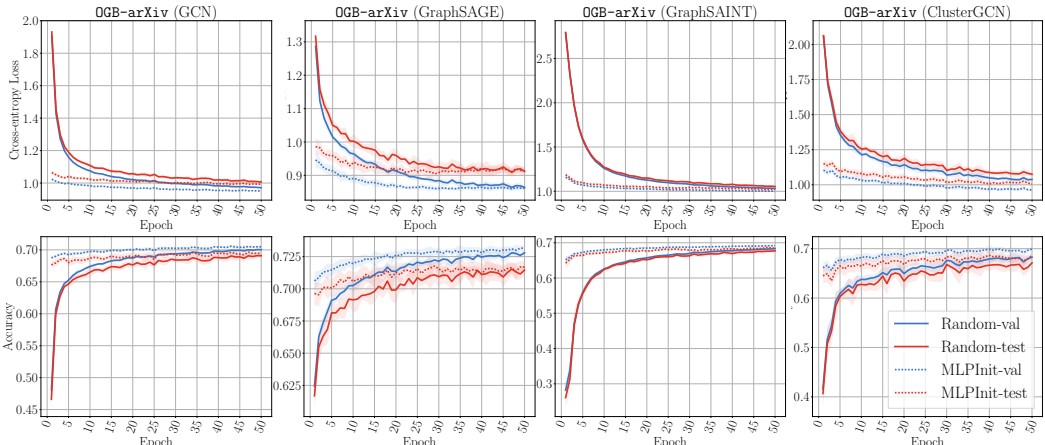

Figure 3: The training curves of different GNNs on OGB-arXiv. GNN with MLPInit generally obtains lower loss and higher accuracy than those with the random initialization and converges faster. The training curves are depicted based on ten runs. More experiment results are in Appendix A.

## 5 EXPERIMENTS

In the next subsections, we conduct and discuss experiments to understand MLPInit from the following aspects: **(i)** training speedup, **(ii)** performance improvements, **(iii)** hyperparameter sensitivity, **(iv)** robustness and loss landscape. For node classification, we consider Flickr, Yelp, Reddit, Reddit2, A-products, and two OGB datasets (Hu et al., 2020), OGB-arXiv and OGB-products as benchmark datasets. We adopt GCN (w/ mini-batch) (Kipf & Welling, 2016a), GraphSAGE (Hamilton et al., 2017), GraphSAINT(Zeng et al., 2019) and ClusterGCN (Chiang et al., 2019) as GNN backbones. The details of datasets and baselines are in Appendices C.1 and C.2, respectively. For the link prediction task, we consider Cora, CiteSeer, PubMed, CoraFull, CS, Physics, A-Photo, and A-Computers as our datasets. Our link prediction setup is using as GCN as an encoder which transforms a graph to node representation $\mathbf{H}$ and an inner-product decoder $\hat{\mathbf{A}} = \text{sigmoid}(\mathbf{H} \cdot \mathbf{H}^T)$ to predict the probability of the link existence, which is discussed in Section 2.

### 5.1 HOW MUCH CAN MLPINIT ACCELERATE GNN TRAINING?

In this section, we compared the training speed of GNNs with random initialization and MLPInit. We computed training epochs needed by GNNs with random initialization to achieve the best test performance. We also compute the running epochs needed by GNNs with MLPInit to achieve comparable test performance. We present the results in Table 3. We also plotted the loss and accuracy curves of different GNNs on OGB-arXiv in Figure 3. We made the following major observations:

Table 4: Performance improvement when GNN with random initialization and with MLPInit achieve best test performance, respectively. Mean and standard deviation are calculated based on ten runs. The best test performance for the two methods is independently selected based on validation data.

| | Methods | Flickr | Yelp | Reddit | Reddit2 | A-products | OGB-arXiv | OGB-products | Avg. |
|---|---|---|---|---|---|---|---|---|---|
| SAGE | Random | $53.72_{\pm0.16}$ | $63.03_{\pm0.20}$ | $96.50_{\pm0.03}$ | $51.76_{\pm2.53}$ | $77.58_{\pm0.05}$ | $72.00_{\pm0.16}$ | $80.05_{\pm0.35}$ | 70.66 |
| | MLPInit | $53.82_{\pm0.13}$ | $63.93_{\pm0.23}$ | $96.66_{\pm0.04}$ | $89.60_{\pm1.60}$ | $77.74_{\pm0.06}$ | $72.25_{\pm0.30}$ | $80.04_{\pm0.62}$ | 76.29 |
| | Improv. | ↑ 0.19% | ↑ 1.43% | ↑ 0.16% | ↑ 73.09% | ↑ 0.21% | ↑ 0.36% | ↓ 0.01% | ↑ 7.97% |
| SAINT | Random | $51.37_{\pm0.21}$ | $29.42_{\pm1.32}$ | $95.58_{\pm0.07}$ | $36.45_{\pm4.09}$ | $59.31_{\pm0.12}$ | $67.95_{\pm0.24}$ | $73.80_{\pm0.58}$ | 59.12 |
| | MLPInit | $51.35_{\pm0.10}$ | $43.10_{\pm1.13}$ | $95.64_{\pm0.06}$ | $41.71_{\pm1.25}$ | $68.24_{\pm0.17}$ | $68.80_{\pm0.20}$ | $74.02_{\pm0.19}$ | 63.26 |
| | Improv. | ↓ 0.05% | ↑ 46.47% | ↑ 0.06% | ↑ 14.45% | ↑ 15.06% | ↑ 1.25% | ↑ 0.30% | ↑ 7.00% |
| C-GCN | Random | $49.95_{\pm0.15}$ | $56.39_{\pm0.64}$ | $95.70_{\pm0.06}$ | $53.79_{\pm2.48}$ | $52.74_{\pm0.28}$ | $68.00_{\pm0.59}$ | $78.71_{\pm0.59}$ | 65.04 |
| | MLPInit | $49.96_{\pm0.20}$ | $58.05_{\pm0.56}$ | $96.02_{\pm0.04}$ | $77.77_{\pm1.93}$ | $55.61_{\pm0.17}$ | $69.53_{\pm0.50}$ | $78.48_{\pm0.64}$ | 69.34 |
| | Improv. | ↑ 0.02% | ↑ 2.94% | ↑ 0.33% | ↑ 44.60% | ↑ 5.45% | ↑ 2.26% | ↓ 0.30% | ↑ 6.61% |
| GCN | Random | $50.90_{\pm0.12}$ | $40.08_{\pm0.15}$ | $92.78_{\pm0.11}$ | $27.87_{\pm3.45}$ | $36.35_{\pm0.15}$ | $70.25_{\pm0.22}$ | $77.08_{\pm0.26}$ | 56.47 |
| | MLPInit | $51.16_{\pm0.20}$ | $40.83_{\pm0.27}$ | $91.40_{\pm0.20}$ | $80.37_{\pm2.61}$ | $39.70_{\pm0.11}$ | $70.35_{\pm0.34}$ | $76.85_{\pm0.34}$ | 64.38 |
| | Improv. | ↑ 0.51% | ↑ 1.87% | ↓ 1.49% | ↑ 188.42% | ↑ 9.22% | ↑ 0.14% | ↓ 0.29% | ↑ 14.00% |

Table 5: The performance of link prediction task. The results are based on ten runs. The experiments on other datasets are presented in Table 6. More experiments are presented in Appendix A.1.

| | Methods | AUC | AP | Hits@10 | Hits@20 | Hits@50 | Hits@100 |
|---|---|---|---|---|---|---|---|
| PubMed | $\text{MLP}_{random}$ | $94.76_{\pm0.30}$ | $94.28_{\pm0.36}$ | $14.68_{\pm2.60}$ | $24.01_{\pm3.04}$ | $40.02_{\pm2.75}$ | $54.85_{\pm2.03}$ |
| | $\text{GNN}_{random}$ | $96.66_{\pm0.29}$ | $96.78_{\pm0.31}$ | $28.38_{\pm6.11}$ | $42.55_{\pm4.83}$ | $60.62_{\pm4.29}$ | $75.14_{\pm3.00}$ |
| | $\text{GNN}_{mlpinit}$ | $97.31_{\pm0.19}$ | $97.53_{\pm0.21}$ | $37.58_{\pm7.52}$ | $51.83_{\pm7.62}$ | $70.57_{\pm3.12}$ | $81.42_{\pm1.52}$ |
| | Improvement | ↑ 0.68% | ↑ 0.77% | ↑ 32.43% | ↑ 21.80% | ↑ 16.42% | ↑ 8.36% |
| DBLP | $\text{MLP}_{random}$ | $95.20_{\pm0.18}$ | $95.53_{\pm0.25}$ | $28.70_{\pm3.73}$ | $39.22_{\pm4.13}$ | $53.36_{\pm3.81}$ | $64.83_{\pm1.95}$ |
| | $\text{GNN}_{random}$ | $96.29_{\pm0.20}$ | $96.64_{\pm0.23}$ | $36.55_{\pm4.08}$ | $43.13_{\pm2.85}$ | $59.98_{\pm2.43}$ | $71.57_{\pm1.00}$ |
| | $\text{GNN}_{mlpinit}$ | $96.67_{\pm0.13}$ | $97.09_{\pm0.14}$ | $40.84_{\pm7.34}$ | $53.72_{\pm4.25}$ | $67.99_{\pm2.85}$ | $77.76_{\pm1.20}$ |
| | Improvement | ↑ 0.39% | ↑ 0.47% | ↑ 11.73% | ↑ 24.57% | ↑ 13.34% | ↑ 8.65% |
| A-Photo | $\text{MLP}_{random}$ | $86.18_{\pm1.41}$ | $85.37_{\pm1.24}$ | $4.36_{\pm1.14}$ | $6.96_{\pm1.28}$ | $12.20_{\pm1.24}$ | $17.91_{\pm1.26}$ |
| | $\text{GNN}_{random}$ | $92.07_{\pm2.14}$ | $91.52_{\pm2.08}$ | $9.63_{\pm1.58}$ | $12.82_{\pm1.72}$ | $20.90_{\pm1.90}$ | $29.08_{\pm2.53}$ |
| | $\text{GNN}_{mlpinit}$ | $93.99_{\pm0.58}$ | $93.32_{\pm0.60}$ | $9.17_{\pm2.12}$ | $13.12_{\pm2.11}$ | $22.93_{\pm2.56}$ | $32.37_{\pm1.89}$ |
| | Improvement | ↑ 2.08% | ↑ 1.97% | ↓ 4.75% | ↑ 2.28% | ↑ 9.73% | ↑ 11.32% |
| Physics | $\text{MLP}_{random}$ | $96.26_{\pm0.11}$ | $95.63_{\pm0.15}$ | $5.38_{\pm1.32}$ | $8.76_{\pm1.37}$ | $15.86_{\pm0.81}$ | $24.70_{\pm1.11}$ |
| | $\text{GNN}_{random}$ | $95.84_{\pm0.13}$ | $95.38_{\pm0.15}$ | $6.62_{\pm1.00}$ | $10.39_{\pm1.04}$ | $18.55_{\pm1.60}$ | $26.88_{\pm1.95}$ |
| | $\text{GNN}_{mlpinit}$ | $96.89_{\pm0.07}$ | $96.55_{\pm0.11}$ | $8.05_{\pm1.44}$ | $13.06_{\pm1.94}$ | $22.38_{\pm1.94}$ | $32.31_{\pm1.43}$ |
| | Improvement | ↑ 1.10% | ↑ 1.22% | ↑ 21.63% | ↑ 25.76% | ↑ 20.63% | ↑ 20.20% |
| | Avg. | ↑ 1.05% | ↑ 1.10% | ↑ 17.81% | ↑ 20.97% | ↑ 14.88% | ↑ 10.46% |

**Observation 3: MLPInit can significantly reduce the training time of GNNs.** In this experiment, we summarize the epochs needed by GNN with random initialization to obtain the best performance, and then we calculate the epochs needed by GNN with MLPInit to reach a comparable performance on par with the randomly initialized GNN. We present the time speedup of MLPInit in Table 3. Table 3 shows MLPInit speed up the training of GNNs by $2-5$ times generally and in some cases even more than 30 times. The consistent reduction of training epochs on different datasets demonstrates that MLPInit can generally speed up GNN training quite significantly.

## 5.2 HOW WELL DOES MLPINIT PERFORM ON NODE CLASSIFICATION AND LINK PREDICTION TASKS?

In this section, we conducted experiments to show the superiority of the proposed method in terms of final, converged GNN model performance on node classification and link prediction tasks. The reported test performances of both random initialization and MLPInit are selected based on the validation data. We present the performance improvement of MLPInit compared to random initialization in Tables 4 and 5 for node classification and link prediction, respectively.

**Observation 4: MLPInit improves the prediction performance for both node classification and link prediction task in most cases.** Table 4 shows our proposed method gains 7.97%, 7.00%, 6.61% and 14.00% improvements for GraphSAGE, GraphSAINT, ClusterGCN, and GCN on average cross

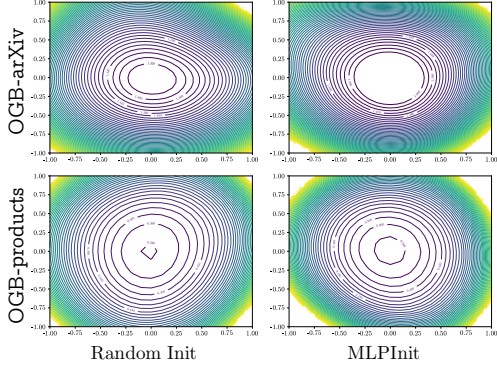

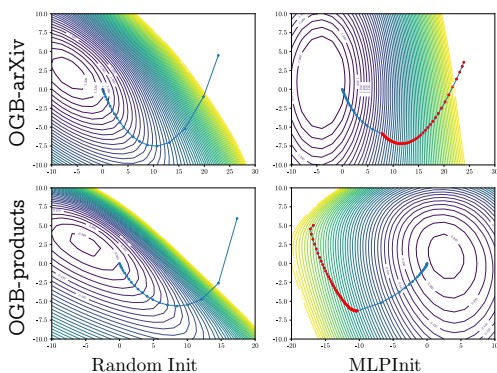

Figure 5: The loss landscape of GNN trained with random initialization (left) and MLPInit (right). The low-loss area of GNNs with MLPInit is larger than that with random initialization.

Figure 6: The training trajectory of the GNN with random initialization (left) and MLPInit (right). The first-phase training of GNNs can be taken over by lightweight MLPs.

all the datasets for the node classification task. The results in Table 5 and Table 6 show our proposed method gains 1.05%, 1.10%, 17.81%, 20.97%, 14.88%,10.46% on average cross various metrics for the link prediction task.

## 5.3 IS MLPINIT ROBUST UNDER DIFFERENT HYPERPARAMETERS?

In practice, one of the most time-consuming parts of training large-scale GNNs is hyperparameter tuning (You et al., 2020a). Here, we perform experiments to investigate the sensitivity of MLPInit to various hyperparameters, including the architecture hyperparameters and training hyperparameters.

**Observation 5: MLPInit makes GNNs less sensitive to hyperparameters and improves the overall performance across various hyperparameters.** In this experiment, we trained PeerMLP and GNN with different "Training HP" (Learning rate, weight decay, and batch size) and "Architecture HP" (i.e., layers, number of hidden neurons), and we presented the learning curves of GNN with different hyperparameters in Figure 4. One can see from the results that GNNs trained with MLPInit have a much smaller standard deviation than those trained with random initialization. Moreover, MLPInit consistently outperforms random initialization in task performance. This advantage allows our approach to saving time in searching for architectural hyperparameters. In practice, different datasets require different hyperparameters. Using our proposed method, we can generally choose random hyperparameters and obtain reasonable and relatively stable performance owing to the PeerMLP's lower sensitivity to hyperparameters.

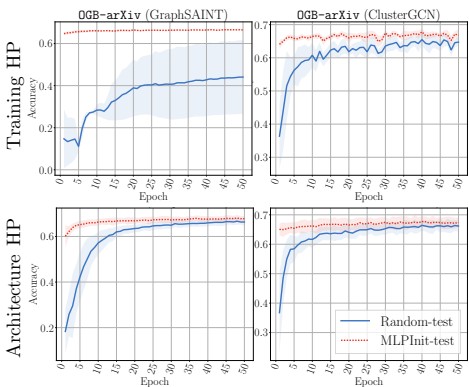

Figure 4: Training curves of GNNs with various hyperparameters. GNNs with MLPInit consistently outperform random initialization and have a smaller standard deviation.

## 5.4 WILL MLPINIT FACILITATE BETTER CONVERGENCE FOR GNNS?

In this experiment, we performed experiments to analyze the convergence of a fine-tuned GNN model. In other words, does this pre-training actually help find better local minima?

**Observation 6: MLPInit finds larger low-loss area in loss landscape for GNNs.** The geometry of loss surfaces reflects the properties of the learned model, which provides various insights to assess the generalization ability (Fort et al., 2019; Huang et al., 2019; Wen et al., 2018) and robustness (Fort et al., 2019; Liu et al., 2020; Engstrom et al., 2019) of a neural network. In this experiment, we plot the loss landscape of GNN (GraphSAGE) with random weight initialization and MLPInit using the visualization tool introduced in Li et al. (2018a). The loss landscapes are plotted based on the training loss and with `OGB-arXiv` and `OGB-products` datasets. The loss landscapes are shown in

Figure 5. For the fair comparison, two random directions of the loss landscape are the same and the lowest values of losses in the landscape are the same within one dataset, 1.360 for `OGB-arXiv` and 0.340 for `OGB-products`. From the loss landscape, we can see that the low-loss area of MLPInit is larger in terms of the same level of loss in each dataset's (row's) plots, indicating the loss landscape of the model trained with MLPInit results in larger low-loss area than the model with random initialization. In summary, MLPInit helps larger low-loss areas in loss landscape for GNNs.

**Observation 7: MLPInit speeds up the optimization process** for GNNs. To further understand the training procedure of MLPInit, we visualize the training trajectories along with loss landscapes through tools (Li et al., 2018a) in Figure 6. In this experiment, we use GraphSAGE on `OGB-arXiv` and `OGB-products` datasets, we first train a PeerMLP and then use the weights of the converged PeerMLP to initialize the GraphSAGE and fine-tune it. Then we plot the training trajectories of PeerMLP and GraphSAGE on the loss landscape of GraphSAGE. The red line indicates the training trajectories of the training of PeerMLP and the blue line indicates the fine-tuning of GNNs. We can see that the end point of the training of MLP (red line) is close to the minima area in the loss landscape. The training trajectory clearly shows the reason why MLPInit works, i.e., the first-phase training of GNNs can be taken over by lightweight MLPs.

## 6 RELATED WORK

In this section, we present several lines of related work and discuss their difference and relation to our work. Also appearing at the same conference as this work, (Yang et al., 2023) concurrently found a similar phonnomenia as our findings and provided a theoretical analysis of it.

**Message passing-based GNNs.** Graph neural networks typically follow the message passing mechanism, which aggregates the information from node's neighbors and learns node representation for downstream tasks. Following the pioneering work GCN (Kipf & Welling, 2016a), several other works (Veličković et al., 2018a; Xu et al., 2018; Balcilar et al., 2021; Thorpe et al., 2021; Brody et al., 2021; Tailor et al., 2021) seek to improve or better understand message passing-based GNNs.

**Efficient GNNs.** In order to scale GNNs to large-scale graphs, the efficiency of GNNs has attracted considerable recent attention. Subgraph sampling technique (Hamilton et al., 2017; Chiang et al., 2019; Zeng et al., 2019) has been proposed for efficient mini-batch training for large-scale graphs. All of these methods follow the message passing mechanism for efficient GNN training. There is another line of work on scalable GNNs, which uses MLPs to simplify GNNs (Zhang et al., 2022; Wu et al., 2019; Frasca et al., 2020; Sun et al., 2021; Huang et al., 2020; Hu et al., 2021). These methods aim to decouple the feature aggregation and transformation operations to avoid excessive, expensive aggregation operations. We compared our work with this line of work in Section 4.2. There are also other acceleration methods, which leverage weight quantization and graph sparsification (Cai et al., 2020). However, these kinds of methods often sacrifice prediction accuracy and increase modeling complexity, while sometimes meriting significant additional engineering efforts.

**GNN Pre-training.** The recent GNN pretraining methods mainly adopt contrastive learning (Hassani & Khasahmadi, 2020; Qiu et al., 2020; Zhu et al., 2020b; 2021; You et al., 2021; 2020c; Jin et al., 2021a; Han et al., 2022; Zhu et al., 2020a). GNN pretraining methods typically leverage graph augmentation to pretrain weights of GNNs or obtain the node representation for downstream tasks. For example, Zhu et al. (2020b) maximizes the agreement between two views of one graph. GNN pre-training methods not only use graph information for model training but also involve extra graph data augmentation operations, which require additional training time and engineering effort.

## 7 CONCLUSION

This work presents a simple yet effective initialization method, MLPInit, to accelerate the training of GNNs, which adopts the weights from their converged PeerMLP initialize GNN and then fine-tune GNNs. With comprehensive experimental evidence, we demonstrate the superiority of our proposed method on training speedup (up to 33× speedup on `OGB-products`), downstream task performance improvements(up to 7.97% performance improvement for GraphSAGE), and robustness improvements (larger minimal area in loss landscape) on the resulting GNNs. Notably, our proposed method is easy to implement and employ in real applications to speed up the training of GNNs.

## ACKNOWLEDGEMENT

We thank the anonymous reviewers for their constructive suggestions and fruitful discussion. Xiaotian would like to thank Zirui Liu from Rice University for the discussion about the training time for MLP and GNN. He would also like to thank Kaixiong Zhou from Rice University and Keyu Duan from National University of Singapore for the discussion about the GNN training on large-scale graphs. Xiaotian would also like to thank Hanqing Zen from Meta Inc., for his valuable feedback and suggestions on the manuscript of this work. We thank Jingyuan Li from the Department of Electrical and Computer Engineering at the University of Washington for identifying a typo in our paper. Portions of this research were conducted with the advanced computing resources provided by Texas A&M High Performance Research Computing. This work is, in part, supported by NSF IIS-1750074 and IIS-1900990. The views and conclusions contained in this paper are those of the authors and should not be interpreted as representing any funding agencies.

## REPRODUCIBILITY STATEMENT

To ensure the reproducibility of our experiments and benefit the research community, we provide the source code at https://github.com/snap-research/MLPInit-for-GNNs. The hyper-parameters and other variables required to reproduce our experiments are described in Appendix D.

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

CONTENTS OF APPENDIX

# A    ADDITIONAL EXPERIMENTS

In this appendix, we present additional experiments to show the superiority of our proposed method, MLPInit, including additional results for link prediction and training curves. We also present more analysis on the weights distribution changes of MLPInit.

## A.1    ADDITIONAL EXPERIMENTAL LINK PREDICTION

Here, we present the additional experiment results on the link prediction task in Table 6, which is similar to our results in Table 5, but on more datasets. In general, we observe that GNN with MLPInit outperforms that with random initialization on the link prediction tasks on these additional datasets. MLPInit does not gain better performance on A-products dataset. We conjecture that the reason for this is that node features may contain less task-relevant information on A-products.

Table 6: The performance of link prediction task. The results are based on ten runs.

| | Methods | AUC | AP | Hits@10 | Hits@20 | Hits@50 | Hits@100 |
|---|---|---|---|---|---|---|---|
| Cora | $\mathrm{MLP_{random}}$ | $91.87_{\pm 1.25}$ | $92.16_{\pm 0.99}$ | $46.98_{\pm 4.33}$ | $58.05_{\pm 6.44}$ | $76.03_{\pm 4.17}$ | $86.28_{\pm 3.05}$ |
| | $\mathrm{GNN_{random}}$ | $91.80_{\pm 1.39}$ | $92.68_{\pm 1.28}$ | $51.48_{\pm 7.57}$ | $63.04_{\pm 6.12}$ | $78.12_{\pm 4.06}$ | $86.07_{\pm 2.77}$ |
| | $\mathrm{GNN_{mlpinit}}$ | $92.93_{\pm 0.88}$ | $93.36_{\pm 0.86}$ | $53.72_{\pm 7.43}$ | $68.65_{\pm 4.22}$ | $80.30_{\pm 2.35}$ | $89.37_{\pm 2.21}$ |
| | Improvement | ↑ 1.23% | ↑ 0.74% | ↑ 4.35% | ↑ 8.91% | ↑ 2.79% | ↑ 3.84% |
| CiteSeer | $\mathrm{MLP_{random}}$ | $90.10_{\pm 0.99}$ | $90.65_{\pm 0.98}$ | $46.64_{\pm 5.11}$ | $57.21_{\pm 4.84}$ | $71.45_{\pm 3.15}$ | $81.93_{\pm 2.24}$ |
| | $\mathrm{GNN_{random}}$ | $89.86_{\pm 1.18}$ | $90.88_{\pm 0.99}$ | $49.98_{\pm 5.54}$ | $58.02_{\pm 5.00}$ | $71.65_{\pm 3.50}$ | $82.11_{\pm 2.84}$ |
| | $\mathrm{GNN_{mlpinit}}$ | $90.51_{\pm 1.07}$ | $90.96_{\pm 0.80}$ | $50.02_{\pm 3.09}$ | $58.77_{\pm 4.48}$ | $71.78_{\pm 4.12}$ | $82.66_{\pm 2.53}$ |
| | Improvement | ↑ 0.73% | ↑ 0.08% | ↑ 0.09% | ↑ 1.29% | ↑ 0.18% | ↑ 0.67% |
| CS | $\mathrm{MLP_{random}}$ | $96.29_{\pm 0.12}$ | $95.79_{\pm 0.13}$ | $13.36_{\pm 1.49}$ | $19.67_{\pm 2.21}$ | $33.46_{\pm 2.17}$ | $46.82_{\pm 1.91}$ |
| | $\mathrm{GNN_{random}}$ | $96.11_{\pm 0.08}$ | $95.75_{\pm 0.10}$ | $14.27_{\pm 2.77}$ | $22.57_{\pm 2.52}$ | $35.40_{\pm 2.01}$ | $48.21_{\pm 2.00}$ |
| | $\mathrm{GNN_{mlpinit}}$ | $96.72_{\pm 0.10}$ | $96.49_{\pm 0.14}$ | $16.96_{\pm 3.37}$ | $25.44_{\pm 3.00}$ | $40.69_{\pm 2.99}$ | $53.78_{\pm 2.00}$ |
| | Improvement | ↑ 0.63% | ↑ 0.77% | ↑ 18.81% | ↑ 12.70% | ↑ 14.96% | ↑ 11.55% |
| A-Computers | $\mathrm{MLP_{random}}$ | $81.85_{\pm 0.79}$ | $82.41_{\pm 0.69}$ | $2.10_{\pm 0.48}$ | $4.13_{\pm 0.86}$ | $7.83_{\pm 0.95}$ | $12.18_{\pm 1.01}$ |
| | $\mathrm{GNN_{random}}$ | $91.78_{\pm 0.48}$ | $91.94_{\pm 0.42}$ | $7.60_{\pm 1.47}$ | $11.10_{\pm 1.74}$ | $18.64_{\pm 1.94}$ | $25.42_{\pm 2.15}$ |
| | $\mathrm{GNN_{mlpinit}}$ | $90.76_{\pm 1.61}$ | $91.06_{\pm 1.47}$ | $6.76_{\pm 3.27}$ | $11.11_{\pm 1.82}$ | $17.40_{\pm 2.58}$ | $24.59_{\pm 2.56}$ |
| | Improvement | ↓ 1.11% | ↓ 0.96% | ↓ 11.04% | ↑ 0.16% | ↓ 6.65% | ↓ 3.26% |
| CoraFull | $\mathrm{MLP_{random}}$ | $95.72_{\pm 0.18}$ | $95.55_{\pm 0.23}$ | $19.38_{\pm 4.71}$ | $27.83_{\pm 3.27}$ | $42.98_{\pm 2.01}$ | $57.20_{\pm 1.27}$ |
| | $\mathrm{GNN_{random}}$ | $95.87_{\pm 0.36}$ | $95.77_{\pm 0.42}$ | $21.33_{\pm 4.77}$ | $30.57_{\pm 3.49}$ | $45.08_{\pm 3.46}$ | $59.58_{\pm 2.53}$ |
| | $\mathrm{GNN_{mlpinit}}$ | $96.71_{\pm 0.16}$ | $96.73_{\pm 0.22}$ | $25.78_{\pm 4.92}$ | $36.68_{\pm 5.36}$ | $53.81_{\pm 2.34}$ | $66.73_{\pm 1.96}$ |
| | Improvement | ↑ 0.87% | ↑ 1.01% | ↑ 20.87% | ↑ 19.98% | ↑ 19.37% | ↑ 12.01% |

## A.2    ADDITIONAL HYPERPARAMETER SENSITIVITY

In this appendix, we present the additional results to explore the sensitivity to the various hyperparameters. The results are the full version of Figure 7.

## A.3    ADDITIONAL TRAINING CURVES

In this appendix, we present the additional training curves of other datasets in Figure 8 and Figure 9, which are additional experimental results of Figure 3. The results comprehensively show the training curves on various datasets. As we can see from Figure 8 and Figure 9 that MLPInit consistently outperforms the random initialization and is able to accelerate the training of GNNs.

## A.4    ADDITIONAL LOSS/ACCURACY CURVES OF PEERMLP AND GNN

In this appendix, we plotted the loss and accuracy curves of PeerMLP and GNN on training/validation/test set and presented the results in Figure 10, which are the additional experimental results to

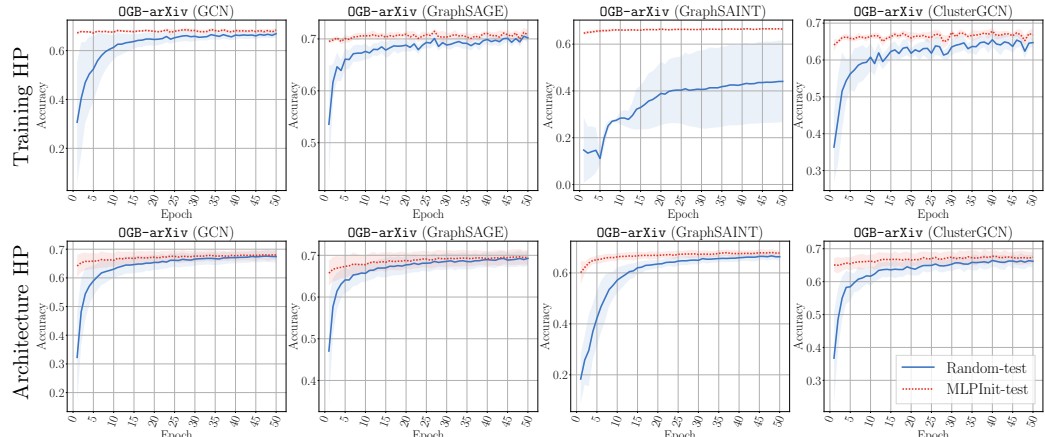

Figure 7: The training curves of the GNNs with different hyperparameters on OGB-arXiv dataset. The training curves of GNN with MLPInit generally obtain lower loss and higher accuracy than those with the random initialization and converge faster. The training curves are depicted based on ten runs.

Figure 2. The results surprisingly show that GNN using the weight from trained PeerMLP has worse cross-entropy loss but better prediction accuracy than PeerMLP. The reason would be that GNN can smooth the prediction logit, making loss worse but accuracy better.

## A.5 TRAINING CURVES OF LINK PREDICTION TASK

To further investigate the training process of the link prediction task, we present the training curves for the link prediction task for each metric we used. The metrics we used are AUC, AP, Hits@K, which are commonly used to evaluate the performance of link prediction (Zhang & Chen, 2018; Zhang et al., 2021a; Zhao et al., 2022b). AUC and AP measure binary classification, reflecting the value of loss for link prediction, and Hits@K is the count of how many positive samples are ranked in the top-K positions against a bunch of negative samples.

The results show that the AUC and AP are easy to train, while the Hits@K is harder to train. The GNN with random initialization needs much more time than MLPInit to obtain a good Hits@K. Since Hits@K is a more realistic metric, the results demonstrate the superiority of our method in the link prediction task. These experimental results also show that node features are important for link prediction task since we can obtain a good performance only with MLP, therefore, MLPInit is beneficial for link prediction since the MLPInit used node feature information only to train PeerMLP.

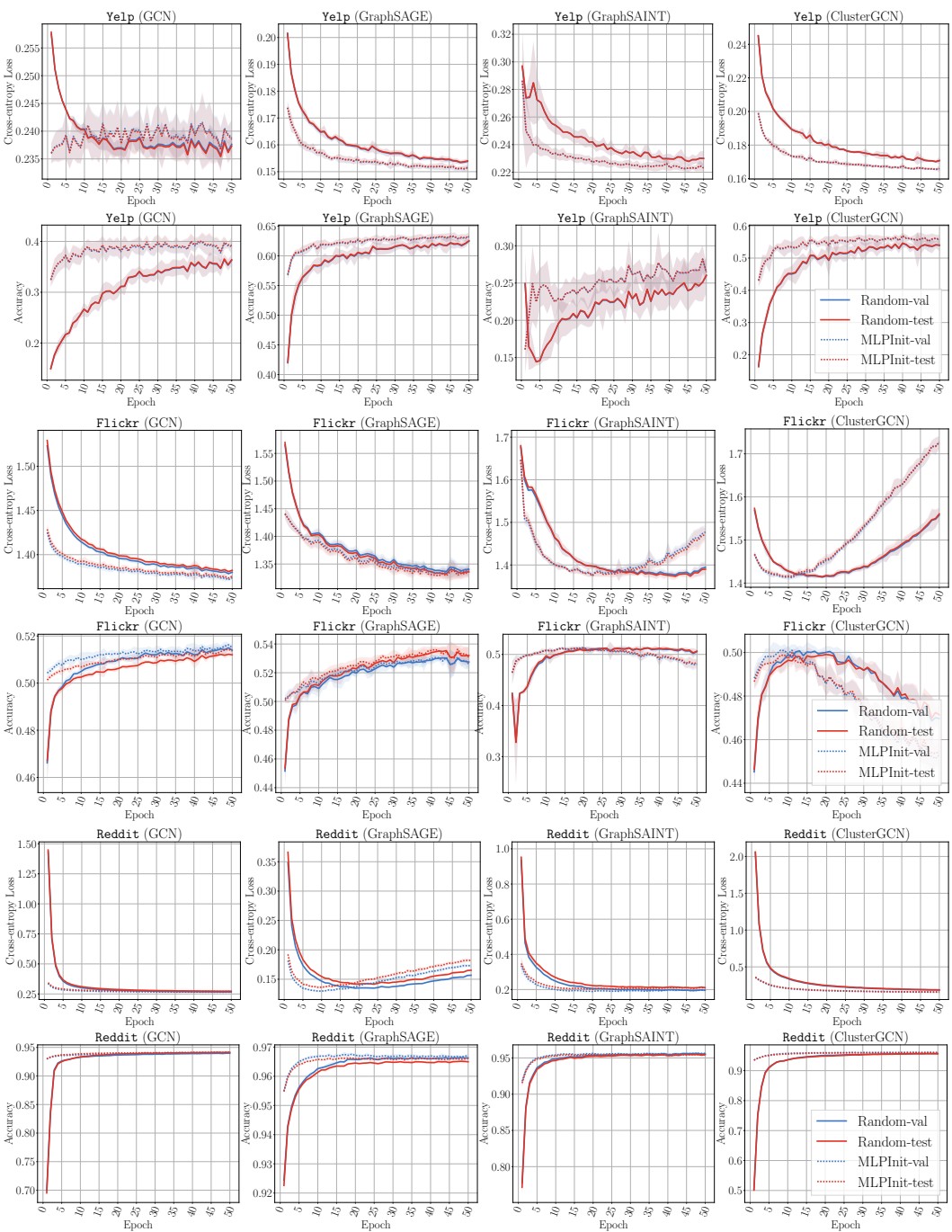

Figure 8: The training curves of GNN with random initialization and MLPInit on Yelp, Flickr, Reddit datasets.

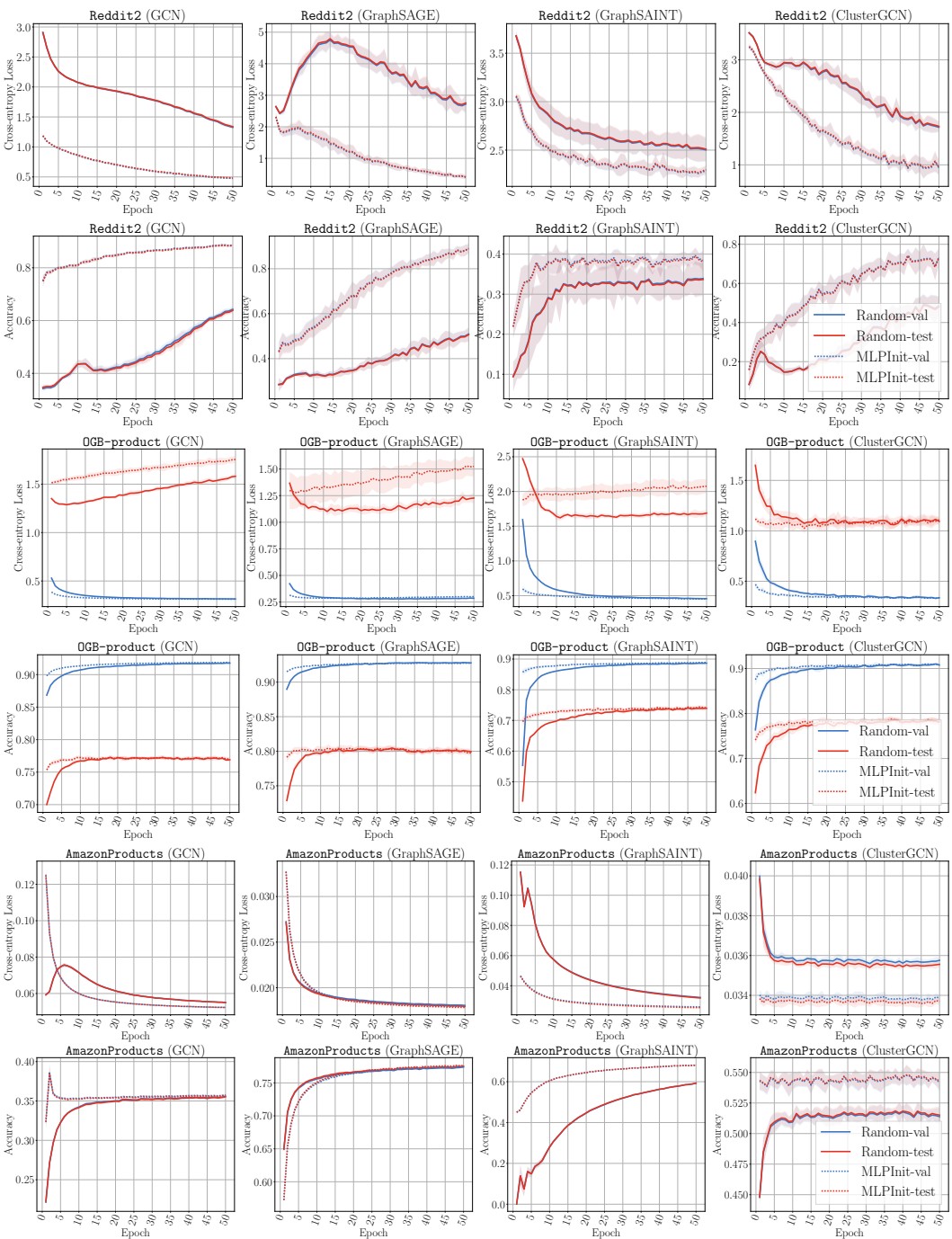

Figure 9: The training curves of GNN with random initialization and with MLPInit on `Reddit2`, `OGB-products`, `A-products` datasets.

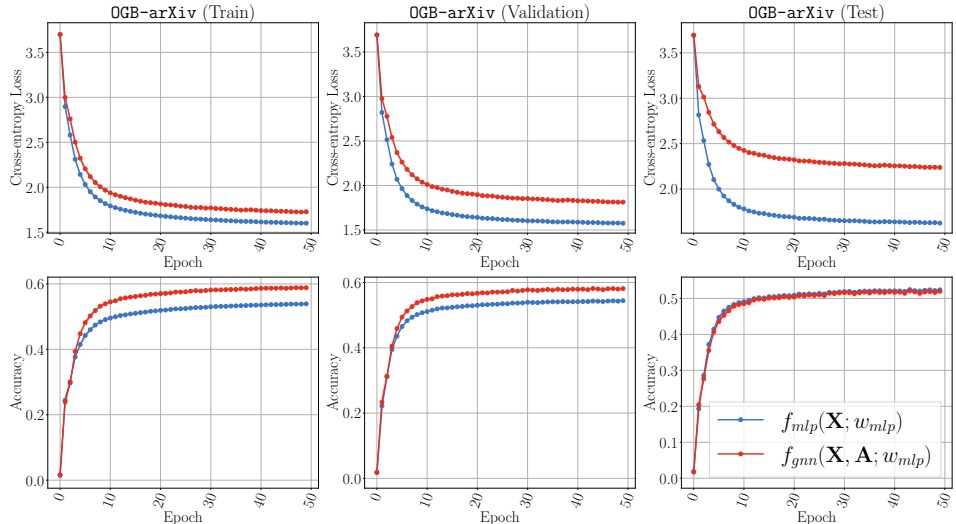

Figure 10: The relation of GNN and MLP during the training of PeerMLP. We report the cross-entropy loss and accuracy on the training/validation/test sets.

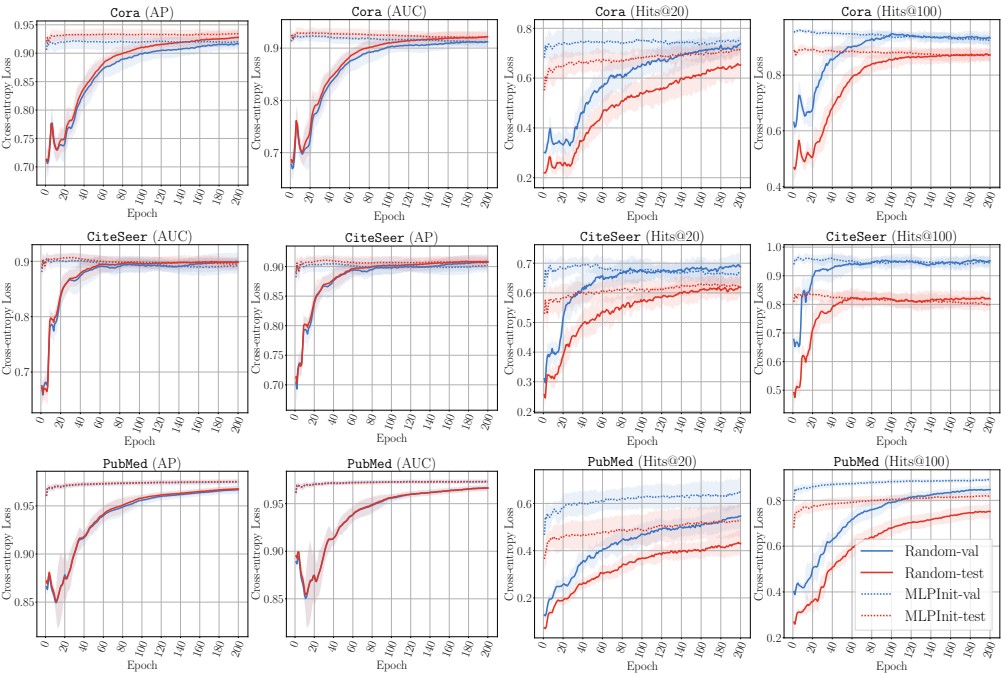

Figure 11: The training curves for link prediction task on Cora, CiteSeer, PubMed datasets. GNN with MLPInit generally obtains higher metrics for link prediction task than those with random initialization and converge faster.

# B    MORE EXPERIMENTS

In this appendix, we conducted additional experiments to further analyze our proposed method MLPInit. We conducted experiments to show that training MLP is much cheaper than training GNN. The results show that the running time of MLP can be negligible compared to that of GNNs. We compare the efficiency of our method to GNN pre-training methods. The comparison to GNN pre-training methods demonstrates the superiority of MLPInit in effectiveness and efficiency. We conduct experiments to investigate the effectiveness of MLPInit on GNN with more complicated aggregators. We provided the new form of our results in Table 4 to show the performance improvements for graph sampling methods and GNN architectures separately.

## B.1    THE PERFORMANCE OF GCN WITH THE WEIGHT OF PEERMLP

In this experiment, we aimed to verify *Observation 2: Converged weights from PeerMLP provide a good GNN initialization* from Section 4 by evaluating the performance of GCN with the weight of PeerMLP on the Cora, CiteSeer, and PubMed datasets. We trained the PeerMLP for GCN, and then calculated the accuracy of the GCN using the well-trained PeerMLP weights (without fine-tuning). We used the public split for these three datasets in a semi-supervised setting. The evaluated models are described in detail below and the results are presented in Table 7.

- PeerMLP: the well-trained PeerMLP for GCN on different datasets.

- GCN w/ $w_{\mathsf{peermlp}}$: the GCN with the weight of well-trained PeerMLP, without fine-tuning.

- GCN: the well-trained graph convolutional neural network.

Table 7: The performance of GNNs and its PeerMLP with the weights of a converged PeerMLP on test data. The accuracy are in percentage (%).

|          | PeerMLP | GCN w/ $w_{\mathsf{peermlp}}$ | Improv. | GCN |
|----------|---------|-------------------------------|---------|------|
| Cora     | 58.50   | 77.60                         | ↑ 32.64% | 82.60 |
| CiteSeer | 60.50   | 69.70                         | ↑ 15.20% | 71.60 |
| PubMed   | 73.60   | 78.10                         | ↑ 6.11%  | 79.80 |

The results show that the GNN w/ $w_{\mathsf{peermlp}}$ significantly outperforms the PeerMLP, even though the weight $w_{\mathsf{peermlp}}$ is trained on PeerMLP. The performance improvements are notable, with increases of $32.64\%$, $15.20\%$, and $6.11\%$ on Cora, CiteSeer, and PubMed datasets, respectively. The results would be additional evidence for *Observation 2: Converged weights from PeerMLP provide a good GNN initialization.*

Moreover, this intriguing phenomenon implies a relationship between MLPs and GNNs that could potentially shed light on the generalization capabilities of GNNs. We believe that our findings will be of significant interest to researchers and practitioners in the field of graph neural networks, and we hope that our work will inspire follow-up research to further explore the relationship between MLPs and GNNs.

## B.2    WEIGHT DIFFERENCE OF GNNS WITH RANDOM INITIALIZATION AND MLPINIT

Prior work (Li et al., 2018a) suggests that "small weights still appear more sensitive to perturbations, and produce sharper looking minimizers." To this end, we explore the distribution of weights of GNNs with both random initialization and MLPInit, and present the results in Figure 12. We can observe that with the same number of training epochs, the weights of GraphSAGE with MLPInit produce more high-magnitude (both positive and negative) weights, indicating the MLPInit can help the optimization of GNN. This difference stems from a straightforward reason: MLPInit provides a good initialization for GNNs since the weights are trained by the PeerMLP before (also aligning with our observations in Section 5.4).

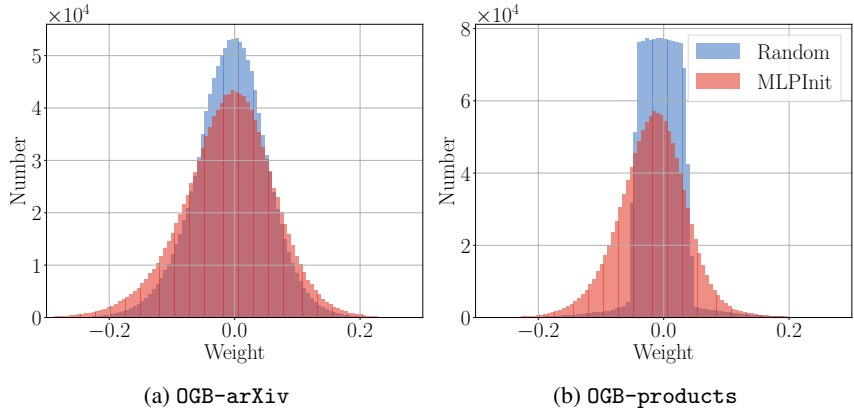

(a) `OGB-arXiv`                (b) `OGB-products`

Figure 12: Weight histograms of GraphSAGE model weights with random initialization (blue) and MLPInit (red) on `OGB-arXiv` (left) and `OGB-products` (right). The results are plotted based on 20 epochs after training GraphSAGE with zero weight decay. Note that MLPInit produces more weights of higher magnitude.

## B.3 Running time comparison of MLP and GNN

We conducted experiments to compare the running time of MLP and GNN (GraphSAGE in this experiment) and presented the running time needed by training MLP and GraphSAGE for one epoch in Table 8. In this experiment, MLP is trained in a full-batch way, and training node features are stored in GPU memory. We adopted the official example code of GraphSAGE [4]. The running time of MLP only needs $1/147$ and $1/2303$ of that of GraphSAGE on `OGB-arXiv` and `OGB-products` datasets. The results show that training MLP is much cheaper than training GNN. In practice, MLP usually only needs to be trained less than 50 epochs to converge. Thus, the training time of MLP in MLPInit is negligible compared to the training time of GNNs.

Table 8: The comparison of the running time of MLP and GraphSAGE. We report the running time of one epoch for MLP, GraphSAGE, and their ratio. The results show that training MLP is much cheaper than training GrapSAGE, especially for the large graph.

| Dataset | MLP | GraphSAGE | MLP/GraphSAGE Ratio |
|---|---|---|---|
| `OGB-arXiv` | $0.035_{\pm 0.000}$ | $5.170_{\pm 0.313}$ | $1/147$ |
| `OGB-products` | $0.076_{\pm 0.000}$ | $175.758_{\pm 9.560}$ | $1/2303$ |

## B.4 Comparison to GNN pre-training methods

In this appendix, we compare the efficiency of our method to GNN pre-training methods. In this experiment, we adopt DGI (Veličković et al., 2018b) as the pre-training method to pretrain the weight of GNN. DGI maximizes the mutual information between patch representations and corresponding high-level summaries of graphs. Since the output of DGI is a hidden representation, we leverage DGI to pretrain weights of the GNN except for the last layer (classification head). We report the final prediction performance of GNN with MLPInit and DGI in Table 9 and reported the running time of MLPInit and DGI in Table 10. The experimental results show that MLPInit obtains rank 1.28 and 1.42 on GraphSAGE and GCN, demonstrating MLPInit outperforms DGI slightly. This might be because the additional classification head for DGI is not pretrained. It is worth noting that DGI is much more time-consuming than MLPInit, as Table 10 show that MLPInit only needs $1/6.59$ and $1/1017.41$ running time of DGI. The comparison to GNN pre-training methods demonstrates the superiority of MLPInit in effectiveness and efficiency.

---

[4] https://github.com/pyg-team/pytorch_geometric/blob/2.0.4/examples/ogbn_products_sage.py

Table 9: The comparison of the performance of MLPInit and DGI. The best performance is in **bold-face**. The Avg.Rank is the average performance rank over 7 datasets.

| | Methods | Flickr | Yelp | Reddit | Reddit2 | A-products | OGB-arXiv | OGB-products | Avg. Rank |
|---|---|---|---|---|---|---|---|---|---|
| SAGE | Random | $53.72_{\pm0.16}$ | $63.03_{\pm0.20}$ | $96.50_{\pm0.03}$ | $51.76_{\pm2.53}$ | $77.58_{\pm0.05}$ | $72.00_{\pm0.16}$ | $80.05_{\pm0.35}$ | 3.00 |
| SAGE | DGI | $\mathbf{53.97}_{\pm0.13}$ | $62.53_{\pm0.31}$ | $96.57_{\pm0.03}$ | $54.82_{\pm1.42}$ | $77.11_{\pm0.08}$ | $71.86_{\pm0.33}$ | $\mathbf{80.24}_{\pm0.57}$ | 1.71 |
| SAGE | MLPInit | $53.82_{\pm0.13}$ | $\mathbf{63.93}_{\pm0.23}$ | $\mathbf{96.66}_{\pm0.04}$ | $\mathbf{89.60}_{\pm1.60}$ | $\mathbf{77.74}_{\pm0.06}$ | $\mathbf{72.25}_{\pm0.30}$ | $80.04_{\pm0.62}$ | 1.28 |
| GCN | Random | $50.90_{\pm0.12}$ | $40.08_{\pm0.15}$ | $92.78_{\pm0.11}$ | $27.87_{\pm3.45}$ | $36.35_{\pm0.15}$ | $70.25_{\pm0.22}$ | $77.08_{\pm0.26}$ | 3.00 |
| GCN | DGI | $\mathbf{51.23}_{\pm0.07}$ | $38.24_{\pm0.54}$ | $\mathbf{94.14}_{\pm0.02}$ | $66.98_{\pm1.22}$ | $35.54_{\pm0.05}$ | $69.40_{\pm0.35}$ | $\mathbf{77.15}_{\pm0.21}$ | 1.57 |
| GCN | MLPInit | $51.16_{\pm0.20}$ | $\mathbf{40.83}_{\pm0.27}$ | $91.40_{\pm0.20}$ | $\mathbf{80.37}_{\pm2.61}$ | $\mathbf{39.70}_{\pm0.11}$ | $\mathbf{70.35}_{\pm0.34}$ | $76.85_{\pm0.34}$ | 1.42 |

Table 10: The comparison of the running time of MLPInit and DGI. We report the running time of one epoch for MLPInit and DGI in the pretraining stage and their ratio. The results show that training MLP is much cheaper than DGI, especially for the large graph.

| Dataset | MLPInit(ours) | DGI | MLPInit/DGI Ratio |
|---|---|---|---|
| OGB-arXiv | $0.035_{\pm0.000}$ | $4.794_{\pm0.055}$ | $1/137$ |
| OGB-products | $0.076_{\pm0.000}$ | $1892.386_{\pm46.176}$ | $1/24798$ |

## B.5 EXPERIMENTS ON MORE COMPLICATED AGGREGATORS

Information aggregators play a vital role in graph neural networks, and recent work proposed complicated aggregators to improve the performance of graph neural networks. In this appendix, we conducted experiments to investigate the effectiveness of MLPInit on GNN with more complicated aggregators. We explore the acceleration effect and prediction accuracy improvement of MLPInit on GNN with more complicated aggregators. The adopted aggregators include Mean, Sum, Max, Median, and Softmax. Their details are presented as follows:

- **Mean** is a commonly used aggregation operator that averages features across neighbors.

- **Max, Median** (Corso et al., 2020) are aggregation operators that take the feature-wise maximum/median across neighbors. The mathematical expressions of Max, Median are $\max_{\mathbf{x}_i \in \mathcal{X}} \mathbf{x}_i$. $\mathrm{median}_{\mathbf{x}_i \in \mathcal{X}} \mathbf{x}_i$.

- **Softmax** (Li et al., 2020) is a learnable aggregation operator, which normalizes the features of neighbors based on a learnable temperature term, as $\mathrm{softmax}(\mathcal{X}|t) = \sum_{\mathbf{x}_i \in \mathcal{X}} \frac{\exp(t \cdot \mathbf{x}_i)}{\sum_{\mathbf{x}_j \in \mathcal{X}} \exp(t \cdot \mathbf{x}_j)} \cdot \mathbf{x}_i$ where $t$ controls the softness of the softmax when aggregating neighbors' features.

We reported the performance improvement, training speedup, and training curves in Table 11, Table 12 and Figure 13. Generally, MLPInit is effective for other aggregators. MLPInit speed up the training of GNNs by $2.06\times$ over four aggregators. Moreover, MLPInit improves the prediction performance by $0.75\%$ over four aggregators. The training curves in Figure 13 show that GNN with MLPInit generally obtain lower loss and higher accuracy than those with random initialization and converge faster.

Table 12: Speed improvement when MLPInit achieves comparable performance with a randomly initialized GNN using various information aggregators. The number reported is the number of training epochs needed.

| | Methods | Mean | Max | Median | Softmax | Avg. |
|---|---|---|---|---|---|---|
| OGB-arXiv | Random | 46.7 | 37.1 | 40.9 | 42.0 | 41.6 |
| OGB-arXiv | MLPInit | 22.7 | 22.4 | 27.2 | 8.8 | 20.2 |
| OGB-arXiv | Improv. | $2.06\times$ | $1.66\times$ | $1.50\times$ | $4.77\times$ | $2.06\times$ |

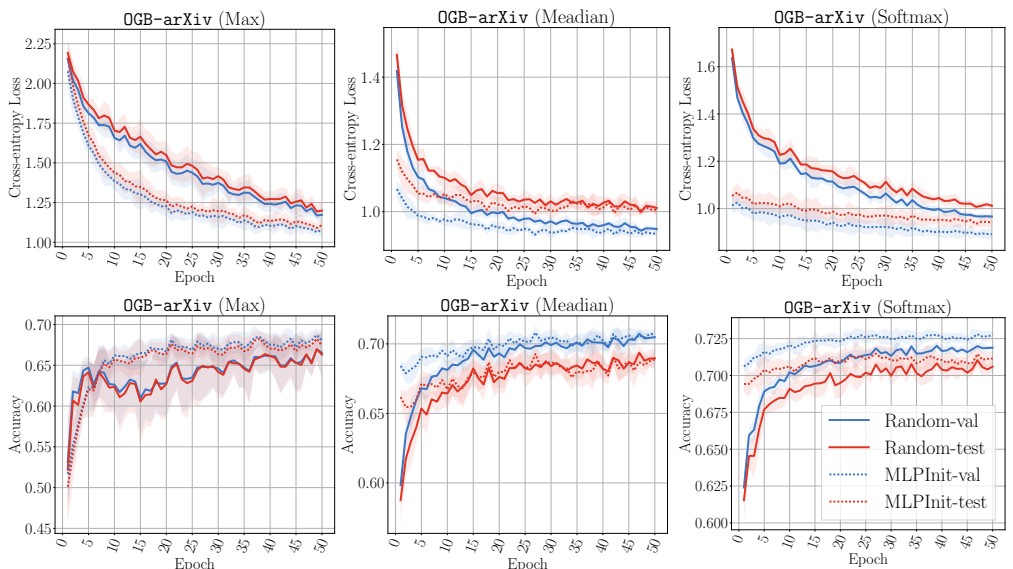

Figure 13: The training curves for GraphSAGE using different aggregators with random initialization and MLPInit.

Table 11: Performance improvement when GNN (GraphSAGE) uses different aggregators with random initialization. Mean, and standard deviation are calculated based on ten runs.

| | Methods | Mean | Max | Median | Softmax | Avg. |
|---|---|---|---|---|---|---|
| OGB-arXiv | Random | $72.00_{\pm0.16}$ | $68.31_{\pm1.00}$ | $69.97_{\pm0.29}$ | $71.05_{\pm0.20}$ | 70.33 |
| | MLPInit | $72.25_{\pm0.30}$ | $69.30_{\pm0.56}$ | $69.95_{\pm0.36}$ | $71.94_{\pm0.18}$ | 70.86 |
| | Improv. | ↑ 0.36% | ↑ 1.44% | ↓ 0.02% | ↑ 1.25% | ↑ 0.75% |

## B.6 RESULTS ON GRAPH SAMPLING METHODS AND GNN ARCHITECTURES

In this appendix, we provided the new form of our results in Table 4 to show the performance improvements for graph sampling methods and GNN architectures separately in Tables 13 and 14.

From the new form of our results, we observed that 1) MLPInit improves the performance of different graph sampling methods as it improves the performance of GraphSAGE, GraphSAINT, and ClusterGCN by 7.96%, 7.00% and 6.62%. 2) MLPInit improves the performance of different graph neural network layers, as it improves the performance of SAGEConv, GCNConv by 6.62%, 14.00%.

## B.7 EXPERIMENTS ON DATASETS WHERE NODE FEATURES ARE LESS IMPORTANT

Our proposed method does depend on a tendency towards node feature-label correlation. Thus, it would likely suffer if features provided less or no information about labels. In this appendix, we conducted experiments on synthetic graphs. The synthetic graphs have differing degrees of correlation between node features and labels. We generated the synthetic graph node features $\mathbf{X}_{synthetic}$ by mixing the original features and random features for each node in the graph as follows:

$$\mathbf{X}_{synthetic} = \lambda\mathbf{X}_{original} + (1 - \lambda)\mathbf{X}_{random}$$

where $\mathbf{X}_{original}$ and $\mathbf{X}_{random}$ are the original features of OGB-arXiv and random features, and $\lambda$ mediates the two. $\lambda$ shows different levels of association between node features and labels. When $\lambda = 0$, the synthetic node features will be the original features of OGB-arXiv. When $\lambda = 1$, the synthetic node features will be completely random features, which are totally uncorrelated to the node labels. We change the value of $\lambda$ to explore the behavior of MLPInit. Note that we initially

Table 13: Performance improvements for various graph sampling methods. The improvement is the overall accuracy performance improvement across all the datasets.

| Sampling | Methods | Flickr | Yelp | Reddit | Reddit2 | A-products | OGB-arXiv | OGB-products | **Improv.** |
|---|---|---|---|---|---|---|---|---|---|
| GraphSAGE | Random | 53.72 | 63.03 | 96.50 | 51.76 | 77.58 | 72.00 | 80.05 | ↑7.96% |
|  | MLPInit | 53.82 | 63.93 | 96.66 | 89.60 | 77.74 | 72.25 | 80.04 |  |
| GraphSAINT | Random | 51.37 | 29.42 | 95.58 | 36.45 | 59.31 | 67.95 | 73.80 | ↑7.00% |
|  | MLPInit | 51.35 | 43.10 | 95.64 | 41.71 | 68.24 | 68.80 | 74.02 |  |
| Cluster-GCN | Random | 49.95 | 56.39 | 95.70 | 53.79 | 52.74 | 68.00 | 78.71 | ↑6.62% |
|  | MLPInit | 49.96 | 58.05 | 96.02 | 77.77 | 55.61 | 69.53 | 78.48 |  |

Table 14: Performance improvements for various GNN architectures. The improvement is the overall accuracy performance improvement across all the datasets.

| GNN layers | Methods | Flickr | Yelp | Reddit | Reddit2 | A-products | OGB-arXiv | OGB-products | **Improv.** |
|---|---|---|---|---|---|---|---|---|---|
| SAGEConv | Random | 49.95 | 56.39 | 95.70 | 53.79 | 52.74 | 68.00 | 78.71 | ↑6.62% |
|  | MLPInit | 49.96 | 58.05 | 96.02 | 77.77 | 55.61 | 69.53 | 78.48 |  |
| GCNConv | Random | 50.90 | 40.08 | 92.78 | 27.87 | 36.35 | 70.25 | 77.08 | ↑14.00% |
|  | MLPInit | 51.16 | 40.83 | 91.40 | 80.37 | 39.70 | 70.35 | 76.85 |  |

conducted the experiments on $\lambda = [0.0, 0.1, 0.2, ..., 1.0]$, and we observed that performance on $\lambda = 0.1$ is much lower than other values, thus we conducted more on $\lambda = [0.05, 0.15]$ around 0.1.

The results shows that

- If node features are uncorrelated to the node labels ($\lambda < 0.2$), GNN with the weights of PeerMLP will not outperform the PeerMLP.

- If the node features are correlated to the node labels ($\lambda > 0.2$), GNN with the weights of PeerMLP will consistently outperform the PeerMLP.

- Overall, MLPInit obtains a better (0.74% average improvement) final accuracy than Random Init over 13 different $\lambda$s.

## B.8 DERIVING THE PEERMLP

In this appendix, we discuss two potential methods to derive the PeerMLP, and discuss their advantages and disadvantages.

### B.8.1 TWO METHODS TO DERIVE THE PEERMLP

The two potential methods are as follows:

1. **Remove the information aggregation operation in GNN**. In this way, we construct a new neural network (PeerMLP, which may contain skip-connections or other complexities of the GNN layer) by entirely removing the neighbor aggregation operation; hence, the trainable weights of PeerMLP will be the same as GNN by design. We need to build a dataloader for it (Algorithm 1). The advantage of this strategy is that it is efficient, since the PeerMLP is a "pure" MLP (no aggregation required by design).

2. **Change the adjacency matrix to an identity matrix** In this way, we use the original GNN architecture, but pretend the set of edges is a set of self-loops on each of the nodes. The advantage of this strategy is that the same dataloader and model structure for GNN can be used for MLP – we don't need to change the input of PeerMLP, which are node features and adjacency matrix (changed to an identity matrix). This facilitates code reuse and ease of engineering/development. However, since we also must use the GNN dataloader and associated model forward operations, we pay for some more training time owing to these operations (graph sampling and identity aggregation).

Table 15: Performance of PeerMLP and GNN with converged weights ($w^*_{mlp}$) of PeerMLP on different $\lambda$ on dataset `OGB-arXiv`. The accuracy in percentage is based on 5 runs.

| $\lambda$ | 0.0 | (0.05) | 0.1 | (0.15) | 0.2 | 0.3 | 0.4 | 0.5 | 0.6 | 0.7 | 0.8 | 0.9 | 1.0 |
|---|---|---|---|---|---|---|---|---|---|---|---|---|---|
| PeerMLP | 5.95 | 6.63 | 11.88 | 16.39 | 19.82 | 23.59 | 25.63 | 31.19 | 40.65 | 48.74 | 53.40 | 55.67 | 56.23 |
| GNN w/ $w^*_{mlp}$ | 5.86 | 5.86 | 5.89 | 8.82 | 24.10 | 32.87 | 40.67 | 51.41 | 55.99 | 59.05 | 61.30 | 62.14 | 62.43 |
| Improv. | −1.53% | −11.59% | −50.43% | −46.16% | 21.58% | 39.37% | 58.71% | 64.81% | 37.73% | 21.15% | 14.79% | 11.63% | 11.03% |

Table 16: Performance of GNN trained with Random Init and MLPInit with different $\lambda$ on `OGB-arXiv`. The accuracy in percentage is the best performance of the two methods and is based on 5 runs.

| $\lambda$ | 0.0 | (0.05) | 0.1 | (0.15) | 0.2 | 0.3 | 0.4 | 0.5 | 0.6 | 0.7 | 0.8 | 0.9 | 1.0 | Avg. |
|---|---|---|---|---|---|---|---|---|---|---|---|---|---|---|
| GNN w/ Random Init | 60.03 | 59.22 | 60.71 | 63.24 | 64.71 | 67.46 | 69.02 | 70.05 | 70.72 | 71.31 | 71.76 | 71.70 | 72.00 | 67.07 |
| GNN w/ MLPInit | 59.78 | 60.47 | 62.53 | 64.49 | 65.94 | 67.84 | 69.11 | 70.47 | 70.70 | 71.45 | 71.50 | 71.92 | 72.25 | 67.57 |
| Improv. | −0.42% | 2.12% | 3.01% | 1.98% | 1.91% | 0.57% | 0.13% | 0.59% | −0.03% | 0.20% | −0.36% | 0.29% | 0.34% | 0.74% |

Table 17: Comparison of the running time of forward and backward for different operations (i.e., $\mathbf{X} \cdot \mathbf{W}$, $\mathbf{A} \cdot \mathbf{Z}$ and $\mathbf{I} \cdot \mathbf{Z}$) in GNNs. The time unit is milliseconds (ms).

| | OGB-arXiv | | | Flickr | | | Yelp | | |
|---|---|---|---|---|---|---|---|---|---|
| #Nodes | 169343 | | | 89250 | | | 716847 | | |
| #Edges | 1166243 | | | 899756 | | | 13954819 | | |
| | Forward | Backward | Total | Forward | Backward | Total | Forward | Backward | Total |
| $\mathbf{Z} = \mathbf{X} \cdot \mathbf{W}$ | 0.56 | 1.31 | 1.87 | 0.51 | 1.80 | 2.31 | 3.10 | 9.14 | 12.24 |
| $\mathbf{H} = \mathbf{A} \cdot \mathbf{Z}$ | 1.47 | 1285.86 | 1287.33 | 1.16 | 842.49 | 843.65 | 13.19 | 16956.19 | 16969.38 |
| $\mathbf{H} = \mathbf{I} \cdot \mathbf{Z}$ | 1.32 | 112.82 | 114.13 | 0.70 | 63.55 | 64.25 | 5.61 | 440.51 | 446.12 |

Table 18: Comparison of the running time of forward and backward for different methods (i.e., $\mathbf{W} \cdot \mathbf{X}$, $\mathbf{A} \cdot \mathbf{W} \cdot \mathbf{X}$ and $\mathbf{I} \cdot \mathbf{W} \cdot \mathbf{X}$). The time unit is milliseconds (ms).

| | OGB-arXiv | | | Flickr | | | Yelp | | |
|---|---|---|---|---|---|---|---|---|---|
| | Forward | Backward | Total | Forward | Backward | Total | Forward | Backward | Total |
| $\mathbf{X} \cdot \mathbf{W}$ | 0.56 | 1.31 | 1.87 | 0.51 | 1.80 | 2.31 | 3.10 | 9.14 | 12.24 |
| $\mathbf{A} \cdot \mathbf{X} \cdot \mathbf{W}$ | 2.03 | 1287.17 | 1289.2 | 1.67 | 844.29 | 845.96 | 16.29 | 16965.33 | 16981.62 |
| $\mathbf{I} \cdot \mathbf{X} \cdot \mathbf{W}$ | 1.88 | 114.13 | 116 | 1.21 | 65.35 | 66.56 | 8.71 | 449.65 | 458.36 |
| ratio of $(\mathbf{X} \cdot \mathbf{W}){:}(\mathbf{A} \cdot \mathbf{X} \cdot \mathbf{W})$ | — | — | 1:689 | — | — | 1:366 | — | — | 1:1387 |
| ratio of $(\mathbf{X} \cdot \mathbf{W}){:}(\mathbf{I} \cdot \mathbf{X} \cdot \mathbf{W})$ | — | — | 1:62 | — | — | 1:28 | — | — | 1:37 |

For more complex graph convolution layers, we take a layer with skip-connection in GNN, $\mathbf{H}^l = \sigma(\mathbf{A}\mathbf{H}^{l-1}\mathbf{W}^l_{gnn} + \mathbf{H}^{l-1})$, as an example. To derive the PeerMLP for the layer with skip-connection, Method 1 directly removes the adjacency matrix $\mathbf{A}$ to yield $\mathbf{H}^l = \sigma(\mathbf{H}^{l-1}\mathbf{W}^l_{gnn} + \mathbf{H}^{l-1})$. Thus, the PeerMLP will also contain a skip-connection operation. Method 2 can be easily and directly adopted since it just alters the adjacency matrix in a trivial way. $\mathbf{H}^l = \sigma(\mathbf{H}^{l-1} \cdot \mathbf{W}^l_{gnn} + \mathbf{H}^{l-1})$.

### B.8.2 DISCUSSION ABOUT THE EFFICIENCY OF PEERMLP DERIVING

Firstly, Method 1 and Method 2 are mathematically equivalent.

Secondly, in the sense of producing computational graphs, they are different. In the next, we only consider the terms $\mathbf{H}^{l-1} \cdot \mathbf{W}^l_{gnn}$ and $\mathbf{I} \cdot \mathbf{H}^{l-1} \cdot \mathbf{W}^l_{gnn}$ in Method 1 and 2 since the rest terms are the same. For operation $\mathbf{H}^{l-1} \cdot \mathbf{W}^l_{gnn}$ in Method 1, it only has one dense matrix multiplication. For operation $\mathbf{I} \cdot \mathbf{H}^{l-1} \cdot \mathbf{W}^l_{gnn}$ in Method 2, it has two steps, one is dense matrix multiplication ($\mathbf{Z} = \mathbf{H}^{l-1} \cdot \mathbf{W}^l_{gnn}$), the other is a sparse matrix multiplication ($\mathbf{I} \cdot \mathbf{Z}$). Thus they will produce different computational graphs (using torch_geometric). Typically, sparse matrix multiplication needs much

more time than dense matrix multiplication (in the case that the sparse matrix is an identity matrix, this still holds if the computation package has no special optimization for the identity matrix).

To investigate the efficiency of these two methods in production environments, we conducted experiments to present the running time of different operations ($\mathbf{X} \cdot \mathbf{W}$, $\mathbf{A} \cdot \mathbf{Z}$ and $\mathbf{I} \cdot \mathbf{Z}$) and different methods ($\mathbf{X} \cdot \mathbf{W}$, $\mathbf{A} \cdot \mathbf{X} \cdot \mathbf{W}$ and $\mathbf{I} \cdot \mathbf{X} \cdot \mathbf{W}$). $\mathbf{X} \cdot \mathbf{W}$ is dense matrix multiplication, $\mathbf{A} \cdot \mathbf{Z}$ and $\mathbf{I} \cdot \mathbf{Z}$ are sparse matrix multiplication. The experiments are conducted with an NVIDIA RTX A5000. The softwares and their version in this experiment are cudatoolkit (11.0.221), PyTorch (1.9.1), torch-sparse (0.6.12). The results are presented in Tables 17 and 18. From the results, we have the following observation:

- Table 17 shows that $\mathbf{I} \cdot \mathbf{Z}$ (sparse matrix multiplication) takes much more time than $\mathbf{X} \cdot \mathbf{W}$ (dense matrix multiplication).

- Table 18 shows that $\mathbf{I} \cdot \mathbf{X} \cdot \mathbf{W}$ takes much more time than $\mathbf{X} \cdot \mathbf{W}$, where the ratio of the running time for ($\mathbf{X} \cdot \mathbf{W}$):($\mathbf{I} \cdot \mathbf{X} \cdot \mathbf{W}$) are $1 : 62$, $1 : 28$, and $1 : 37$ for `OGB-arXiv`, `Flickr` and `Yelp`, respectively. The results indicate that the $\mathbf{I} \cdot \mathbf{X} \cdot \mathbf{W}$ is not equivalent to $\mathbf{X} \cdot \mathbf{W}$ in the sense of producing computational graph (at least for the sparse matrix multiplication in torch-sparse package).

- The choice between the two ultimately boils down to the intended setting. Method 2 is easier for development, and Method 1 is more optimized for speed (and thus useful in resource-constrained settings, or for efficiency in production environments).

## C    DATASETS AND BASELINES

In the appendix, we present the details of datasets and baselines for node classification and link prediction tasks.

### C.1    DATASETS FOR NODE CLASSIFICATION

The details of datasets used for node classification are listed as follows:

- `Yelp` (Zeng et al., 2019) contains customer reviewers as nodes and their friendship as edges. The node features are the low-dimensional review representations for their reviews.

- `Flickr` (Zeng et al., 2019) contains customer reviewers as nodes and their common properties as edges. The node features the 500-dimensional bag-of-word representation of the images.

- `Reddit`, `Reddit2` (Hamilton et al., 2017) is constructed by Reddit posts. The node in this dataset is a post belonging to different communities. `Reddit2` is the sparser version of `Reddit` by deleting some edges.

- `A-products` (Zeng et al., 2019) contains products and its categories.

- `OGB-arXiv` (Hu et al., 2020) is the citation network between all arXiv papers. Each node denotes a paper and each edge denotes citation between two papers. The node features are the average 128-dimensional word vector of its title and abstract.

- `OGB-products` (Hu et al., 2020; Chiang et al., 2019) is Amazon product co-purchasing network. Nodes represent products in Amazon, and edges between two products indicate that the products are purchased together. Node features in this dataset are low-dimensional representations of the product description text.

We present the statistics of datasets used for node classification task in Table 19.

Table 19: Statistics for datasets used for node classification.

| Dataset | # Nodes. | # Edges | # Classes | # Feat | Density |
|---|---|---|---|---|---|
| Flickr | 89,250 | 899,756 | 7 | 500 | 0.11‰ |
| Yelp | 716,847 | 13,954,819 | 100 | 300 | 0.03‰ |
| Reddit | 232,965 | 114,615,892 | 41 | 602 | 2.11‰ |
| Reddit2 | 232,965 | 23,213,838 | 41 | 602 | 0.43‰ |
| A-products | 1,569,960 | 264,339,468 | 107 | 200 | 0.11‰ |
| OGB-arXiv | 169,343 | 1,166,243 | 40 | 128 | 0.04‰ |
| OGB-products | 2,449,029 | 61,859,140 | 47 | 218 | 0.01‰ |

### C.2    BASELINES FOR NODE CLASSIFICATION

We present the details of GNN models as follows:

- **GCN** (Kipf & Welling, 2016a; Hamilton et al., 2017) is the original graph convolution network, which aggregates neighbor's information to obtain node representation. In our experiment, we train GCN in a mini-batch fashion by adopting a subgraph sampler from Hamilton et al. (2017).

- **GraphSAGE** (Hamilton et al., 2017) proposes a graph sampling-based training strategy to scale up graph neural networks. It samples a fixed number of neighbors per node and trains the GNNs in a mini-batch fashion.

- **GraphSAINT** (Zeng et al., 2019) is a graph sampling-based method to train GNNs on large-scale graphs, which proposes a set of graph sampling algorithms to partition graph data into subgraphs. This method also presents normalization techniques to eliminate biases during graph sampling.

- **ClusterGCN** (Chiang et al., 2019) is proposed to train GCNs in small batches by using the graph clustering structure. This approach samples the nodes associated with the dense subgraphs identified by the graph clustering algorithm. Then the GCN is trained by the subgraphs.

## C.3 DATASETS FOR LINK PREDICITON

For link prediction task, we consider `Cora`, `CiteSeer`, `PubMed`, `CoraFull`, `CS`, `Physics`, `A-Photo`. and `A-Computers` as our baselines. The details of the datasets used for node classification are listed as follows:

- `Cora`, `CiteSeer`, `PubMed` (Yang et al., 2016) are representative citation network datasets. These datasets contain a number of research papers, where nodes and edges denote documents and citation relationships, respectively. Node features are low-dimension representations for papers. Labels indicate the research field of documents.
- `CoraFull` (Bojchevski & Günnemann, 2018) is a citation network that contains papers and their citation relationship. Labels are generated based on topics. This dataset is the original data of the entire network of Cora, and Cora dataset in Planetoid is its subset.
- `CS`, `Physics` (Shchur et al., 2018) are from Co-author dataset, which is co-authorship graph based on the Microsoft Academic Graph from the KDD Cup 2016 challenge. Nodes in this dataset are authors, and edges indicate co-author relationships. Node features represent paper keywords. Labels indicate the research field of the authors.
- `A-Photo`, `A-Computers` (Shchur et al., 2018) are two datasets from Amazon co-purchase dataset (McAuley et al., 2015). Nodes in this dataset represent products, while edges represent the co-purchase relationship between two products. Node features are low-dimension representations of product reviews. Labels are categories of products.

We also present the statistics of datasets used for link prediction task in Table 20.

Table 20: Statistics for datasets used for link prediction.

| Dataset | # Nodes | # Edges | # Feat | Density |
|---|---|---|---|---|
| Cora | 2,708 | 5,278 | 1,433 | 0.72‰ |
| CiteSeer | 3,327 | 4,552 | 3,703 | 0.41‰ |
| PubMed | 19,717 | 44,324 | 500 | 0.11‰ |
| DBLP | 17,716 | 105,734 | 1,639 | 0.34‰ |
| CoraFull | 19,793 | 126,842 | 8,710 | 0.32‰ |
| A-Photo | 7,650 | 238,162 | 745 | 4.07‰ |
| A-Computers | 13,752 | 491,722 | 767 | 2.60‰ |
| CS | 18,333 | 163,788 | 6,805 | 0.49‰ |
| Physics | 34,493 | 495,924 | 8,415 | 0.14‰ |

## C.4 BASELINES FOR LINK PREDICITON

Our link prediction setup is consistent with our discussion in Section 2, in which we use an inner-product decoder $\hat{\mathbf{A}} = \text{sigmoid}(\mathbf{H} \cdot \mathbf{H}^T)$ to predict the probability of the link existence. We presented the results in Tables 5 and 6. Following standard benchmarks (Hu et al., 2020), the evaluation metrics adopted are AUC, Average Precision (AP), are hits ratio (Hit@#). The experimental details of link prediction are presented in Appendix D.5.

# D   IMPLEMENTATION DETAILS

In this appendix, we present the hyperparameters used for the node classification task and link prediction task for all models and datasets.

## D.1   RUNNING ENVIRONMENT

We run our experiments on the machine with one NVIDIA Tesla T4 GPU (16GB memory) and 60GB DDR4 memory to train the models. For `A-products` and `OGB-products` datasets, we run the experiments with one NVIDIA A100 GPU (40GB memory). The code is implemented based on PyTorch 1.9.0 (Paszke et al., 2019) and PyTorch Geometric 2.0.4 (Fey & Lenssen, 2019). The optimizer is Adam (Kingma & Ba, 2015) to train all GNNs and their PeerMLPs.

## D.2   EXPERIMENT SETTING FOR FIGURE 2

In this experiment, we use GraphSAGE as GNN on `OGB-arXiv` dataset. We construct that PeerMLP($f_{mlp}(\mathbf{X}; w_{mlp})$ ) for GraphSAGE ($f_{gnn}(\mathbf{X}, \mathbf{A}; w_{mlp})$) and train it for $50$ epochs. From the mathematical expression, GraphSAGE and its PeerMLP share the same weights $w_{mlp}$ and the weights $w_{mlp}$ are only trained by PeerMLP. We use the trained weights for GraphSAGE to make inference along the training procedure. For the landscape, suppose we have two optimal weight $w_{gnn}^*$ and $w_{mlp}^*$ for GraphSAGE and its PeerMLP, the middle one is the loss landscape based on the PeerMLP with optimal with $w_{mlp}^*$ while the right one is the loss landscape based the GraphSAGE with optimal weights $w_{gnn}^*$.

## D.3   EXPERIMENT SETTING FOR FIGURES 3, 8 AND 9 AND TABLES 3 AND 4

In this appendix, we present the detailed experiment setting for our main result Figures 3, 8 and 9 and Tables 3 and 4. We construct the PeerMLP for each GNN. We first train the PeerMLP for $50$ epochs and save the best model with the best validation performance. And then, we use the weight trained by PeerMLP to initialize the GNNs, then fine-tune the GNNs. To investigate the performance of GNNs, we fine-tune the GNNs for $50$ epochs. We list the hyperparameters used in our experiments. We borrow the optimal hyperparameters from paper (Duan et al., 2022). And our code is also based on the official code [5] of paper (Duan et al., 2022). For datasets not included in paper (Duan et al., 2022), we use the heuristic hyperparameter setting for them.

## D.4   EXPERIMENT SETTING FOR TABLE 2

The GNN used in Table 2 is GraphSAGE. We construct the PeerMLP for GraphSAGE on `OGB-arXiv` and `OGB-products` datasets. We first train the PeerMLP for $50$ epochs and save the best model with the best validation performance. And then we infer the test performance with PeerMLP and GraphSAGE with the weight trained by PeerMLP and we report the test performance in Table 2.

## D.5   EXPERIMENT SETTING FOR TABLES 5 AND 6

In this appendix, we present the detailed experiment setting for the link prediction task. We adopt the default setting for the official examples [6] of PyTorch Geometric 2.0.4. The GNN used for the link prediction task is a 2-layer GCN, and the decoder is the commonly used inner-product decoder as $\hat{\mathbf{A}} = \text{sigmoid}(\mathbf{H} \cdot \mathbf{H}^T)$ (Kipf & Welling, 2016b).

## D.6   EXPERIMENT SETTING FOR FIGURE 4

We explore two kinds of hyperparameters "Training HP" (Learning rate, weight decay, batch size, and dropout) and "Architecture HP" (i.e., layers, number of hidden neurons), in this experiment.

- Training Hyperparameter ( Training HP ), total combinations: $2 \times 2 \times 2 \times 2 = 16$

---

[5] https://github.com/VITA-Group/Large_Scale_GCN_Benchmarking
[6] https://github.com/pyg-team/pytorch_geometric/blob/2.0.4/examples/link_pred.py

Table 21: Training configuration for GNNs training in Figures 3, 8 and 9 and Tables 3 and 4.

| Model | Dataset | #Layers | #Hidden | Learning rate | Batch size | Dropout | Weight decay | Epoch |
|---|---|---|---|---|---|---|---|---|
| GraphSAGE | Flickr | 4 | 512 | 0.0001 | 1000 | 0.5 | 0.0001 | 50 |
| | Yelp | 4 | 512 | 0.0001 | 1000 | 0.2 | 0 | 50 |
| | Reddit | 4 | 512 | 0.0001 | 1000 | 0.2 | 0 | 50 |
| | Reddit2 | 4 | 512 | 0.0001 | 1000 | 0.2 | 0 | 50 |
| | A-products | 4 | 512 | 0.001 | 1000 | 0.5 | 0 | 50 |
| | OGB-arXiv | 4 | 512 | 0.001 | 1000 | 0.5 | 0 | 50 |
| | OGB-products | 4 | 512 | 0.001 | 1000 | 0.5 | 0 | 50 |
| GraphSAINT | Flickr | 4 | 512 | 0.001 | 5000 | 0.2 | 0.0004 | 50 |
| | Yelp | 2 | 128 | 0.01 | 5000 | 0.7 | 0.0002 | 50 |
| | Reddit | 2 | 128 | 0.01 | 5000 | 0.7 | 0.0002 | 50 |
| | Reddit2 | 2 | 128 | 0.01 | 5000 | 0.7 | 0.0002 | 50 |
| | A-products | 2 | 128 | 0.01 | 5000 | 0.2 | 0 | 50 |
| | OGB-arXiv | 2 | 128 | 0.01 | 5000 | 0.2 | 0 | 50 |
| | OGB-products | 2 | 128 | 0.01 | 5000 | 0.2 | 0 | 50 |
| ClusterGCN | Flickr | 2 | 256 | 0.001 | 5000 | 0.2 | 0.0002 | 50 |
| | Yelp | 4 | 256 | 0.0001 | 2000 | 0.5 | 0 | 50 |
| | Reddit | 4 | 256 | 0.0001 | 2000 | 0.5 | 0 | 50 |
| | Reddit2 | 4 | 256 | 0.0001 | 2000 | 0.5 | 0 | 50 |
| | A-products | 4 | 128 | 0.001 | 2000 | 0.2 | 0.0001 | 50 |
| | OGB-arXiv | 4 | 128 | 0.001 | 2000 | 0.2 | 0.0001 | 50 |
| | OGB-products | 4 | 128 | 0.001 | 2000 | 0.2 | 0.0001 | 50 |
| GCN | Flickr | 2 | 512 | 0.0001 | 1000 | 0.5 | 0.0001 | 50 |
| | Yelp | 2 | 512 | 0.0001 | 1000 | 0.2 | 0 | 50 |
| | Reddit | 2 | 512 | 0.0001 | 1000 | 0.2 | 0 | 50 |
| | Reddit2 | 2 | 512 | 0.0001 | 1000 | 0.2 | 0 | 50 |
| | A-products | 2 | 512 | 0.001 | 1000 | 0.5 | 0 | 50 |
| | OGB-arXiv | 2 | 512 | 0.001 | 1000 | 0.5 | 0 | 50 |
| | OGB-products | 2 | 512 | 0.001 | 1000 | 0.5 | 0 | 50 |

- – Learning rate: $\{0.001, 0.0001\}$
- – Weight decay: $\{1e-4, 4e-4\}$
- – Batch size: $\{500, 1000\}$
- – Dropout: $\{0.2, 0.5\}$
- • Architecture Hyperparameter ( Architecture HP ), total combinations: $3 \times 5 = 15$
  - – Number of layers: $\{2, 3, 4\}$
  - – number of hidden neurons: $\{32, 64, 128, 256, 512\}$

In Figures 4 and 7, we plotted the learning curves based on the mean and standard deviation over all the hyperparameters combinations.

