# OpenReview forum: "MLPInit: Embarrassingly Simple GNN Training Acceleration with MLP Initialization"
_ICLR.cc/2023/Conference — ICLR 2023 poster_

### Official Review · Reviewer_hojB · 2022-10-25

**Confidence:** 4
**Correctness:** 3
**Technical Novelty And Significance:** 3
**Empirical Novelty And Significance:** 4
**Recommendation:** 8

**Clarity, Quality, Novelty And Reproducibility:**

Clarity: Good

Quality: Good

Novelty: Good

Reproducibility: Good


**Strength And Weaknesses:**

Strength:

1. The writing of this paper is very clear, the motivation is strong, and the paper is organized in an easy-to-follow structure.
2. The observation of this paper that a fully-converged MLP can be a perfect initialization for GNN training is very interesting and meaningful, and the explanation for this makes sense to me.
3. The proposed MLPInit makes sense and is orthogonal to other GNN acceleration techniques.
4. Experimental results are very impressive, the performance improvement in terms of both acceleration and efficacy is very clear.
5. Example code is provided and the experimental setting is explained, so I think there is no reproducibility concern.


Weakness:

It would be better if the major observation can be explained with some theoretical analysis. I know this might be very challenging, and I agree it is ok if this paper is without any theories, it's just how to further enhance the solidity of this work.

Some questions:
1. In table 2, if I understand correctly, the results in the "GNN" column are without any fine-tuning right? So it is not exactly MLPinit, right? Then comparing these two columns give us the intuition that, the parameters learned with MLP can get further improved by taking the advantage of graph structure with message passing. Then, can we add one more column to show the results of MLPinit in the same table, so that people can get a more straightforward intuition that this MLP-learned parameter can get further improved so we still need to do some training on the GNN?
2. Compare to the line of GNN pre-training work, in terms of just time, then MLPinit has a clear advantage. But if we only consider the efficacy, can MLPInit get comparable (or even better) performance than other GNN pre-training work?

**Summary Of The Paper:**

This work proposes to accelerate GNN training by initializing GNN model with a converged MLP of the same parameter size (referred to as peerMLP in the paper).  The authors first point out their empirical observations that 1) a GNN and its peerMLP have same convergence trend and 2) a converged MLP is not good enough and the performance can be further improved by taking the advantage of GNN's message passing scheme. Then, the authors propose the MLPInit based on their observations and demonstrate its great performance in terms of both efficiency and efficacy with rich experiments.

**Summary Of The Review:**

I really like this paper, it is neat but strong. The observation that a fully-converged MLP can be a perfect initialization for GNN training is very interesting and meaningful, and the experimental results are very impressive. Though there is no theoretical analysis as a mathematical justification, enough empirical results are given to demonstrate the rationale of the proposed MLPInit. Therefore, I think it deserves a clear acceptance.

---

> ### Author Response · Authors · 2022-11-17
> **Response to Reviewer hojB (Part 2/2)**
>
> **Q2: Compare to the line of GNN pre-training work, in terms of just time, then MLPinit has a clear advantage. But if we only consider the efficacy, can MLPInit get comparable (or even better) performance than other GNN pre-training work?**
>
> We thank you for your constructive suggestion. We added the following experiments to compare the performance of MLPInit to a GNN pre-training method, DGI [1]. Since the output of DGI is a hidden representation, we leverage DGI to pretrain weights of the GNN except for the last layer (classification head) The code of DGI is the official sample code given in PyTorch Geometric [2]. The results are as follows (Table 11 in the updated paper):
>
> Table: The comparison of the performance improvement of MLPInit and DGI.
> | GNNs         | Methods  | Flickr          | Yelp              | Reddit            | Reddit2           | A-products        | OGB-arXiv       | OGB-products    | Avg. Rank |
> | --------     | :------- | :----------     | :----------       | :----------       | :----------       | :----------       | :----------     | :-------------  | :---------|
> | GraphSAGE    | Random   | 53.72±0.16      | 63.03±0.20        | 96.50±0.03        | 51.76±2.53        | 77.58±0.05        | 72.00±0.16      | 80.05±0.35      | 3.00      |
> |              | DGI      | **53.97**±0.13  | 62.53±0.31        | 96.57±0.03        | 54.82±1.42        | 77.11±0.08        | 71.86±0.33      | **80.24**+0.57  | 1.71      |
> |              | MLPInit  | 53.82±0.13      | **63.93**±0.23    | **96.66**±0.04    | **89.60**+1.60    | **77.74**±0.06    | **72.25**±0.30  | 80.04+0.62      | 1.28      |
> | GCN          | Random   | 50.90±0.12      | 40.08±0.15        | 92.78±0.11        | 27.87+3.45        | 36.35±0.15        | 70.25±0.22      | 77.08±0.26      | 3.00      |
> |              | DGI      | **51.23**±0.07  | 38.24±0.54        | **94.14**±0.02    | 66.98±1.22        | 35.54±0.05        | 69.40±0.35      | **77.15**±0.21  | 1.57      |
> |              | MLPInit  | 51.16±0.20      | **40.83**±0.27    | 91.40+0.20        | **80.37**±2.61    | **39.70**±0.11    | **70.35**±0.34  | 76.85±0.34      | 1.42      |
>
>
> The experimental results show that MLPInit obtains ranks of $1.28$ and $1.42$ on GraphSAGE and GCN, demonstrating that MLPInit slightly outperforms DGI in terms of efficacy. This might be because the additional classification head for DGI is not pretrained.
>
> Additionally, we compared the running time of DGI and MLPInit pretraining for one epoch. The running time of our method is only $1/137$ and $1/24798$ of DGI for OGB-arxiv and OGB-Products, respectively.  The following results (Table 12 in the updated paper) show that DGI is much more time-consuming than MLPInit.
>
> Table: The comparison of the running time of MLPInit and DGI.
> | Dataset      | MLPInit (ours)    | DGI              |  MLPInit/DGI |
> | --------     | --------          | --------         | --------:     |
> | OGB-arxiv    |0.03502±0.00004    | 4.794±0.055      | **1/137**    |
> | OGB-Products |0.07631±0.00020    | 1892.386±46.176  | **1/24798**  |
>
> [**References**]\
> [1] Velickovic, Petar, William Fedus, William L. Hamilton, Pietro Liò, Yoshua Bengio, and R. Devon Hjelm. "Deep Graph Infomax." ICLR (2019).\
> [2] https://github.com/pyg-team/pytorch_geometric/blob/2.0.4/examples/infomax_inductive.py
>
> Thanks,\
> Authors

---

> ### Author Response · Authors · 2022-11-17
> **Response to Reviewer hojB (Part 1/2)**
>
> Dear Reviewer hojB,
>
> We sincerely appreciate your careful reading of our paper. We are excited that you appreciated the contribution and the novelty of our work, and thank you for your positive assessment. Below we address your questions and comments:
>
> **W1: It would be better if the major observation can be explained with some theoretical analysis. I know this might be very challenging, and I agree it is ok if this paper is without any theories, it's just how to further enhance the solidity of this work.**
>
> Thank you for your insightful suggestion. We completely agree that theoretical analysis would further enhance the solidity of this work. In our current form, we pay more attention to developing a strong empirical understanding and discussing this work in a paper given its large practical implications. We indeed leave the theoretical analysis for future work, and we hope our interesting findings will inspire following work from others to extend this direction.
>
> **Q1: can we add one more column to show the results of MLPinit in the same table, so that people can get a more straightforward intuition that this MLP-learned parameter can get further improved so we still need to do some training on the GNN?**
>
>
> Thank you for the valuable suggestion. We revised the paper by updating Table 2 to the following table:
>
> | Dataset      | Methods    | PeerMLP       | GNN           | MLPInit       | Improv. |
> | --------     | ---------- | :------------ | :------------ | :------------ | ------: |
> | OGB-arxiv    | GraphSAGE  | 56.04±0.27    | 62.87±0.95    | 72.25±0.30    | ↑12.18% |
> |              | GraphSAINT | 53.88±0.41    | 63.26±0.71    | 68.80±0.20    | ↑17.41% |
> |              | ClusterGCN | 54.47±0.41    | 60.81±1.30    | 69.53±0.50    | ↑11.63% |
> |              | GCN        | 56.31±0.21    | 56.28±0.89    | 70.35±0.34    | ↓0.04%  |
> | OGB-Products | GraphSAGE  | 63.43±0.14    | 74.32±1.04    | 80.04±0.62    | ↑17.16% |
> |              | GraphSAINT | 57.29±0.32    | 69.00±1.54    | 74.02±0.19    | ↑20.44% |
> |              | ClusterGCN | 59.53±0.46    | 71.74±0.70    | 78.48±0.64    | ↑20.51% |
> |              | GCN        | 62.63±0.15    | 71.11±0.10    | 76.85±0.34    | ↑13.55% |
>
> As you mentioned, the performance gap between GNN amd MLPInit clearly show that *MLP-learned parameter can get further improved so we still need to do some training on the GNN*.

---

> ### Author Response · Authors · 2022-11-23
> **Looking forward to your response**
>
> Dear Reviewer hojB,
>
> Thank you again for the positive assessment.
>
> As we are nearing the end of the discussion phase, we would like to know if our response has addressed your questions. If you have any further questions, we would be more than happy to address them.
>
> Thank you again for your time. And if our response has addressed your concerns, we hope you will re-evaluate our paper in light of the revised version and our initial response.
>
> Thanks,\
> Authors

---

> > ### Comment · Reviewer_hojB · 2022-11-23
> > **I am very satisfied with the author response**
> >
> > Dear Authors,
> >
> > I checked your response and I have no more other questions or concerns, the response looks good to me and addresses my previous questions well. Good luck with your submission!
> >
> > Thank you!

---

> > > ### Author Response · Authors · 2022-11-23
> > > **We are glad that we have addressed your concerns**
> > >
> > > Dear Reviewer hojB,
> > >
> > > Thank you very much for carefully reading our response and your re-assessment.  We are happy that our initial response has addressed your concerns.
> > >
> > > Best wishes to you!!
> > >
> > > Thanks,\
> > > Authors

---

### Official Review · Reviewer_pot2 · 2022-10-25

**Confidence:** 4
**Correctness:** 4
**Technical Novelty And Significance:** 3
**Empirical Novelty And Significance:** 4
**Recommendation:** 8

**Clarity, Quality, Novelty And Reproducibility:**

Clarity: The paper is sometimes a bit hard to read (see my questions and nitpicks), but overall the ideas are clear.



Quality: The quality of the experimental evaluation seems sufficient.



Reproducibility: The results look reproducible.

**Strength And Weaknesses:**

=== Strengths ===

(S1): The paper proposes a very simple trick to speed up GNN training, and then does a thorough empirical evaluation to show that it actually works. It's very surprising to me that the weights of the MLP actually transfer, and thus I wouldn't be inclined to believe in this method a priori, but the experimental results look sound.

(S2): The paper presents good visualizations to explain why MLPInit could work; e.g. Figure 6 shows an interesting phenomena where the first part of GNN training trajectory is very similar no matter if one trains in GNN-space or MLP-space.

(S3): I imagine the results presented in this work can also lead to some further insights about the inner workings of GNNs. Perhaps GNNs are more similar to MLPs-on-node-features than we think, or (as suggested by Figure 6), a large initial part of their training is approximately equivalent to fitting node features in isolation, and connectivity is taken into account reluctantly, when needed.



=== Weaknesses ===

(W1): I am not convinced by the loss landscape comparison. As far as I understand, for Figure 5 the authors selected the contour lines so that the lowest line (1.360 for OGB-arXiv) is the same in both plots. However, isn't it the case that the MLPInit-based GNN achieves lower loss than the plain one, and thus there would be more contour lines in its plot, but these are just not shown? I think this plot may be conflating flatness (large area with similar/equivalent loss) with having a low minimum loss in general (which also means there is a large area with loss smaller than a given constant, but that area is not flat). I don't think this is a very major result in the paper, yet I think some clarification is needed here.

(W2): The reading experience is a bit bumpy, with many typos or small errors. While this doesn't prevent understanding of the work, it would be nice to clean up these issues. I list some of them in the "Other comments" section, and defer the (many) minor ones to the "Nitpicks" section.



=== Other comments ===

(O1): Comparing Figure 2 with Table 2, is it the case that plugging in the MLP weights into the GNN makes loss worse but accuracy better? If so then that's an interesting finding that could be highlighted.

(O2): "MLP whose weights can be made identical" in the abstract can sound a bit confusing before one reads the paper; maybe expand that part a bit, or mention that this is about the shape of the weights?

(O3): The contribution section mentions that training MLPs is cheaper than training GNNs, but gives no quantitative value for that, leaving the reader wondering (until they get to the actual experiments). Maybe mention the rough ratio there already.

(O4): Do the 0's in Table 4 denote cases where fine-tuning could not improve on top of the MLPInit-initialized weights? Is that not taken into account in the reported averages (which would make sense as technically it's "infinite speedup")?

(O5): The title of Observations 6 and 7 is basically the same?



=== Nitpicks ===

Here I include some final nitpicks; they are here to help improve and polish the paper.

- Figure 1 is hard to read, partially because the grid lines are very thick and interfere with the text.

- Missing dots in a few places (after citations in the into, in Observation 1), missing space (in Observation 2), or extra comma + space (Observation 5).

- "With that MLPs train faster than GNNs in mind" - maybe "Having the fact that MLPs train faster..."

- "we denote the prediction targets denoted by" - repetition

- "with comprehensive empirical analysis" - I would add "a" before "comprehensive"

- In caption of Table 1 "We compute the time used for forward and backward of two operation." - what does "two operation" mean here?

- In Section 4.2, "In the section" -> "In this section".

- Overall there are many typos, grammar errors, or broken sentences, in Section 4.2, beginning of Section 5, and also Sections 6-7. I will not list all of them, but I encourage the authors to carefully proof-read their paper, especially the second half.

**Summary Of The Paper:**

This paper proposes to accelerate the training of GNNs. This is done by defining an MLP limited to just the node features with a matching parameter count/shapes, training that MLP, and then using the resulting weights to initialize the GNN. Surprisingly, this proves to be very effective, as the MLP weights are close to optimal for the GNN.

**Summary Of The Review:**

Overall, while I have a few reservations about one of the results (W1) and the paper is a bit hard to parse at times (W2), the general idea presented in this work is interesting and practical, and the evaluation is thorough. Hence I lean towards acceptance, assuming the authors clean up the text and clarify the things I asked about in the main part of my review.

=== Update after author response ===

I'm happy with the author response, which cleared up many of my concerns. The authors also added several useful results to further strengthen the work empirically. Consequently, I raise my scores on correctness (3 -> 4), empirical novelty (3 -> 4), overall (6 -> 8) and also confidence (3 -> 4).

---

> ### Author Response · Authors · 2022-11-17
> **Response to Reviewer pot2 (Part 2/2)**
>
> **Q5: Do the 0's in Table 4 denote cases where fine-tuning could not improve on top of the MLPInit-initialized weights? Is that not taken into account in the reported averages (which would make sense as technically it's "infinite speedup")?**
>
> Yes, 0s in Table 3 denote cases where fine-tuning could not improve on top of the MLPInit-initialized weights. We revised the numbers in Table 3 by recomputing the reported averages of speedup without the 0s. The new results indicates the speed up for GraphSAINT and GCN are 2.77X and 2.27X, which are still strong speedup. -- thanks for pointing this out!
>
> It is worthy noting that this interesting phenomenon supports our claim that *our method can also directly serve as the final, or deployed GNN model, in resource-constrained settings*. If the GNNs with the optimal weights of PeerMLP consistently outperform PeerMLP, or are even close to the performance of the GNNs, we can use the GNNs with the optimal weights of PeerMLP as the final deployed model.
>
>
>
> **Q6: The title of Observations 6 and 7 is basically the same?**
>
> Thank you for pointing this out. Observations 6 says MLPInit finds larger low-loss area in loss landscape for GNNs, whereas observations 7 claims that MLPInit speeds up the optimization process for GNNs. We have revised our paper to make the claim clear by rephrasing Observation 6 to *MLPInit finds larger low-loss area in loss landscape for GNNs* and rephrasing Observation 7 to *MLPInit speeds up the optimization process for GNNs*.
>
>
> **Q7: Minor comments (nitpicks)**
>
> We sincerely appreciate your careful reading of our paper and your efforts in pointing these missed typos. We addressed them carefully. We also revised our paper to improve its readability by:
> - replotting Figure 1 by drawing thicker lines and removing some text;
> - fixing all the typos you pointed out;
> - carefully proofreading the entire paper.
>
>
> [**References**]\
> [1] https://github.com/pyg-team/pytorch_geometric/blob/2.0.4/examples/ogbn_products_sage.py
>
>
> Thanks,\
> Authors

---

> ### Author Response · Authors · 2022-11-17
> **Response to Reviewer pot2 (Part 1/2)**
>
> Dear Reviewer pot2,
>
> We sincerely thank you for you careful reading and constructive suggestions for our paper. We address your concerns as follows:
>
> **Q1: for Figure 5 the authors selected the contour lines so that the lowest line (1.360 for OGB-arXiv) is the same in both plots. I think this plot may be conflating flatness (large area with similar/equivalent loss) with having a low minimum loss in general.**
>
> Thank you for your insightful comment on the loss landscape. We agree with you that a more precise statement is that MLPInit results in a larger, low-loss area than GNN with random initialization. We revised the paper accordingly (See Section 5.4).
>
> **Q2: Comparing Figure 2 with Table 2, is it the case that plugging in the MLP weights into the GNN makes loss worse but accuracy better? If so then that's an interesting finding that could be highlighted.**
>
> We thank you for your careful reading of our work. To investigate the effect of plugging in the MLP weights into the GNN, we plotted the loss/accuracy curves on training/validation/test data set, respectively (See Figure 11 in Appendix D.2). The experimental results surprisingly show that GNN using the weight from trained PeerMLP has worse cross-entropy loss but better prediction accuracy than PeerMLP. The reason would be that GNN can smooth the prediction logit, which can make loss worse but accuracy better. This counter-intuitive phenomena would intrigue the following work to explore the generalization of GNNs.
>
>
> **Q3: "MLP whose weights can be made identical" in the abstract can sound a bit confusing before one reads the paper; maybe expand that part a bit, or mention that this is about the shape of the weights?**
>
> Thank you for your valuable suggestion. We have revised this sentence to "We observe that for most message passing-based GNNs, we can trivially derive an analog MLP (we call this a PeerMLP) *with an equivalent weight space, by setting the trainable parameters with the same shapes."* to make it more clear.
>
> **Q4: The contribution section mentions that training MLPs is cheaper than training GNNs, but gives no quantitative value for that, leaving the reader wondering (until they get to the actual experiments). Maybe mention the rough ratio there already.**
>
> Thanks for the comment. Table 1 does show these results (namely, that  training MLPs is cheaper than training GNNs). We also present an additional result to further support it.
> * Table 1 made a more precise comparsion for the computional time of MLPs and GNNs. The results in Table 1 show that the running time of $Z=WX$ is much smaller than $H=AZ$. MLP only has the operation $Z=WX$  while GNN has two operations, $Z=WX$ and $H=AZ$. $H=AZ$ needs much more computational time (724X, 665X, 3199X) that $Z=WX$, indicating that the runing time of MLP is negligible compared to that of GNNs.
> * We conducted additional experiments and presented the training time of MLPs/GNNs for one epoch to show that the training time of MLPs is negligible compared to that of GNNs.  In this experiment, MLP is trained in a full-batch way and node features are stroed in GPU memory. We adopt the official example code of GraphSAGE [1] of torch geometric.
>
> Table: Running time comparison of MLP and GraphSAGE (The unit is second. For a fair comparsion, the experiments are conducted on the same computer with NVIDIA RTX A5000.)
> |    Dataset         |    MLP               |    GraphSAGE       |  MLP/GraphSAGE Ratio       |
> | :----------------- | :------------------- | :----------------- | -------------------------: |
> | OGB-arxiv          | 0.03502±0.00004      | 5.170±0.313        | **1/147**                      |
> | OGB-Products       | 0.07631±0.00020      | 175.758±9.560      | **1/2303**                     |

---

> ### Author Response · Authors · 2022-11-23
> **Looking forward to your response**
>
> Dear Reviewer pot2,
>
> Thank you again for the positive assessment.
>
> As we are nearing the end of the discussion phase, we would like to know if our response has addressed your questions. If you have any further questions, we would be more than happy to address them.
>
> Thank you again for your time. And if our response has addressed your concerns, we hope you will raise your score in light of the revised version and our initial response.
>
> Thanks,\
> Authors

---

> > ### Comment · Reviewer_pot2 · 2022-11-26
> > **Response to authors**
> >
> > I've reviewed your responses and I'm happy with them. I also appreciate the breadth of new empirical results added during the rebuttal. I raised my score (6 -> 8) and confidence (3 -> 4) to reflect this.
> >
> > (Side comment: I saw that the paper update introduced a few typos like double dots, please remember to do a final proof-reading to clean up such things)

---

> > > ### Author Response · Authors · 2022-11-27
> > > **Thank you for raising the score!**
> > >
> > > Dear Reviewer pot2,
> > >
> > > We thank you for your reply and for raising the score. We are glad that our response has addressed your concerns. And we will polish the paper according to your suggestions.
> > >
> > > Thanks,\
> > > Authors

---

> > > > ### Comment · Reviewer_pot2 · 2022-11-27
> > > > **Response**
> > > >
> > > > Thanks. One more comment: in private reviewer discussions, we were chatting about how to generally define `PeerMLP` for complex GNNs which involve various layers and operations. My thinking is that we need to construct a `PeerMLP` such that all of the weights of the original GNN are used in some way (otherwise some of the layers would recieve no training at all). One way to define it seems to be the following: take the original GNN, but pretend the set of edges is a set of self-loops on each of the nodes. I think for simple layers this gives the same result as how you define `PeerMLP` in your paper (?), but it takes care of what to do for more complex layers. Of course, for efficient implementation one would have to take advantage of the fact that this fake set of edges is exactly `[(0, 0), (1, 1), ..., (num_verts - 1, num_verts - 1)]` to write out the `PeerMLP` in an efficient way.

---

> > > > > ### Author Response · Authors · 2022-11-28
> > > > > **Thanks for your follow-up comments**
> > > > >
> > > > >
> > > > > Dear Reviewer pot2,
> > > > >
> > > > > Thank you so much for your insightful comments and the ongoing discussion among reviewers. We clarify how to derive the PeerMLP to address your concerns as follows:
> > > > >
> > > > >
> > > > > The two methods to derive the PeerMLP you mentioned are both reasonable, and each has its own benefits (in fact, we discussed both these ways when we were working on this project).
> > > > >
> > > > > 1. **Remove the information aggregation operation in GNN**, as you mentioned, "construct a PeerMLP such that all of the weights of the original GNN are used in some way". In this way, we will construct a new neural network (PeerMLP, which may contain skip-connections or other complexities of the GNN layer) by entirely removing the neighbor aggregation operation; hence, the trainable weights of PeerMLP will be the same as GNN by design. We need to build a dataloader for it (Algorithm 1). The advantage of this strategy is that it is efficient, since the PeerMLP is a "pure" MLP (no aggregation required by design). This implementation is shown in the demo code at [https://anonymous.4open.science/r/mlpinit-628C/demo/ogbn_sage.py](https://anonymous.4open.science/r/mlpinit-628C/demo/ogbn_sage.py).
> > > > >
> > > > > 2. **Change the adjacency matrix to an identity matrix**, as you mentioned, "take the original GNN, but pretend the set of edges is a set of self-loops on each of the nodes."  The advantage of this strategy is that the same dataloader and model structure for GNN can be used for MLP -- we don't need to change the input of PeerMLP, which are node features and adjacency matrix (changed to an identity matrix). This facilitates code reuse and ease of engineering/development.  However, since we also must use the GNN dataloader and associated model forward operations, we pay for some more training time owing to these operations (graph sampling and identity aggregation).
> > > > >
> > > > >
> > > > > For more complex graph convolution layers,
> > > > > - Method 2 can be easily and directly adopted since it just alters the adjacency matrix in a trivial way.
> > > > > - For Method 1, we take the graph convolution with skip-connection as example: The layer with skip-connection in GNN is $\mathbf{H}^{l} = \sigma( \mathbf{A} \mathbf{W}\_{gnn}^{l} \mathbf{H}^{l-1} + \mathbf{H}^{l-1})$. To construct a layer of the PeerMLP, we can also directly remove the adjacency matrix $\mathbf{A}$ to yield $\mathbf{H}^{l} = \sigma( \mathbf{W}\_{gnn}^{l} \mathbf{H}^{l-1} + \mathbf{H}^{l-1})$. Thus, the PeerMLP will also contrain a skip-connection operation.  A corresponding implementation will likely be faster than Method 2, but may require a bit more implementation.
> > > > >
> > > > > The choice between the two ultimately boils down to the intended setting.  The Method 2 is easier for development, and Method 1 more optimized for speed (and thus useful in resource-constrained settings, or for efficiency in production environments).
> > > > >
> > > > >
> > > > > We sincerely thank you again for your insightful comments and the discussion among reviewers. **We will also add a discussion on how to derive the PeerMLP for different graph convolutions to the updated version of our paper.**
> > > > >
> > > > > Thanks,\
> > > > > Authors

---

> > > > > > ### Comment · Reviewer_pot2 · 2022-11-28
> > > > > > **Response**
> > > > > >
> > > > > > Yes, I was thinking in terms of Method 2 mostly for ease of explanation of the idea. Just to clarify: are the two methods equivalent in the sense of producing the same computational graph? In your example of Method 1, you're removing `A`, which is like assuming `A = I`, which Method 2 also does.

---

> > > > > > > ### Author Response · Authors · 2022-11-29
> > > > > > > **Reply: Response [Part 2/2]**
> > > > > > >
> > > > > > > Table1: Comparison of the running time of forward and backward for different operations (i.e., $\mathbf{W} \cdot \mathbf{X}$, $\mathbf{A} \cdot \mathbf{Z}$ and $\mathbf{I} \cdot \mathbf{Z}$) in GNNs. The time unit is milliseconds (ms).
> > > > > > > | Operations                                    |         |     OGB-arxiv       |     |         |  Flickr  |                |          |   Yelp    |                 |
> > > > > > > | :---------                                   | :------:| :-------:| :-------------:| :------:| :-------:|  :-----------: |:-------: |:--------: | :-------------: |
> > > > > > > | #Nodes                                       |         |  169343  |                |         | 89250    |                |          | 716847    |                 |
> > > > > > > | #Edges                                       |         |  1166243 |                |         | 899756   |                |          | 13954819  |                 |
> > > > > > > |                                              | Forward | Backward | Total          | Forward | Backward | Total          | Forward  | Backward  | Total           |
> > > > > > > | $\mathbf{Z} = \mathbf{W} \cdot \mathbf{X}$   | 0.56    | 1.31     | 1.87           | 0.51    | 1.80     | 2.31           | 3.10     | 9.14      | 12.24           |
> > > > > > > | $\mathbf{H} = \mathbf{A} \cdot \mathbf{Z}$   | 1.47    | 1285.86  | 1287.33        | 1.16    | 842.49   | 843.65         | 13.19    | 16956.19  | 16969.38        |
> > > > > > > | $\mathbf{H} = \mathbf{I} \cdot \mathbf{Z}$   | 1.32    | 112.82   | 114.13         | 0.70    | 63.55    | 64.25          | 5.61     | 440.51    | 446.12          |
> > > > > > >
> > > > > > >
> > > > > > > Table2: Comparison of the running time of forward and backward for different methods (i.e., $\mathbf{W} \cdot \mathbf{X}$, $\mathbf{A} \cdot \mathbf{W} \cdot \mathbf{X}$ and $\mathbf{I} \cdot \mathbf{W} \cdot \mathbf{X}$). The time unit is milliseconds (ms).
> > > > > > >
> > > > > > > | Methods                                                                                |         |     OGB-arxiv       |     |         |  Flickr  |                |          |   Yelp    |                 |
> > > > > > > | :---------                                                                             | :------:| :-------:| :-------------:| :------:| :-------:|  :-----------: |:-------: |:--------: | :-------------: |
> > > > > > > |                                                                                        | Forward | Backward | Total          | Forward | Backward | Total          | Forward  | Backward  | Total           |
> > > > > > > | $\mathbf{W} \cdot \mathbf{X}$                                                          | 0.56    | 1.31     | 1.87           | 0.51    | 1.80     | 2.31           | 3.10     | 9.14      | 12.24           |
> > > > > > > | $\mathbf{A} \cdot \mathbf{W} \cdot \mathbf{X}$                                         | 2.03	   | 1287.17	| 1289.2	       | 1.67	   | 844.29	  | 845.96	       | 16.29	  | 16965.33	|16981.62         |
> > > > > > > | $\mathbf{I} \cdot \mathbf{W} \cdot \mathbf{X}$                                         | 1.88    | 114.13 	| 116	           | 1.21	   | 65.35	  | 66.56	         | 8.71	    | 449.65	  | 458.36          |
> > > > > > > | ratio of ($\mathbf{W} \cdot \mathbf{X}$):($\mathbf{A} \cdot \mathbf{W} \cdot \mathbf{X}$) | ---     | ---    	| $1:689$        | --- 	   | ---  	  | $1:366$	       | --- 	    | ---   	  | $1:1387$        |
> > > > > > > | ratio of ($\mathbf{W} \cdot \mathbf{X}$):($\mathbf{I} \cdot \mathbf{W} \cdot \mathbf{X}$) | ---     | ---     	| $1:62$         | ---	   | ---  	  | $1:28$	       | ---	    | ---   	  | $1:37$          |
> > > > > > >
> > > > > > > Note that the numbers in above tables are slightly different than the numbers in Table 1 in our paper since we the results in Table 1 in our paper are obtained with NVIDIA RTX A100 while the above tables are obtained with NVIDIA RTX A5000.
> > > > > > >
> > > > > > >
> > > > > > >
> > > > > > > We observed that
> > > > > > >
> > > > > > > - Table 1 shows that $\mathbf{I} \cdot \mathbf{Z}$ (sparse matrix multiplication)  takes much more time than $\mathbf{W} \cdot \mathbf{X}$ (dense matrix multiplication).
> > > > > > > - Table 2 shows that $\mathbf{I} \cdot \mathbf{W} \cdot \mathbf{X}$ takes much more time than $\mathbf{W} \cdot \mathbf{X}$, where the ratio of the running time for ($\mathbf{W} \cdot \mathbf{X}$):($\mathbf{I} \cdot \mathbf{W} \cdot \mathbf{X}$) are $1:62$, $1:28$, and $1:37$ for OGB-arxiv, Flickr and Yelp, respectively. The results indicate that the $\mathbf{I} \cdot \mathbf{W} \cdot \mathbf{X}$ is not equivalent to $\mathbf{W} \cdot \mathbf{X}$ in the sense of producing computational graph (at least for the sparse matrix multiplication in torch-sparse package).
> > > > > > >
> > > > > > >
> > > > > > > We sincerely thank you again for your follow-up comments. In the updated version of our paper, we will also discuss the efficiency analysis of two approaches to deriving PeerMLP.
> > > > > > >
> > > > > > > Thanks,\
> > > > > > > Authors

---

> > > > > > > ### Author Response · Authors · 2022-11-29
> > > > > > > **Reply: Response [Part 1/2]**
> > > > > > >
> > > > > > > Dear Reviewer pot2,
> > > > > > >
> > > > > > > Thank you for your follow-up questions. We would like to clarify your question about the equivalence of the two methods. Despite the mathematical equivalence of the two methods, the computation graphs of them are not precisely the same, and we show the analysis and impact below.
> > > > > > >
> > > > > > > In our previous examples, we have
> > > > > > >
> > > > > > > - The layer with skip-connection in GNN is $\mathbf{H}^{l} = \sigma( \mathbf{A} \cdot \mathbf{W}\_{gnn}^{l} \cdot \mathbf{H}^{l-1} + \mathbf{H}^{l-1})$.
> > > > > > >
> > > > > > > - Method 1 (remove the adjacency matrix): $\mathbf{H}^{l} = \sigma( \mathbf{W}\_{gnn}^{l} \cdot \mathbf{H}^{l-1} + \mathbf{H}^{l-1})$.
> > > > > > >
> > > > > > > - Method 2 (change to an identity matrix): $\mathbf{H}^{l} = \sigma( \mathbf{I} \cdot \mathbf{W}\_{gnn}^{l} \cdot \mathbf{H}^{l-1} + \mathbf{H}^{l-1})$, where $\mathbf{I}$ is an identity matrix.
> > > > > > >
> > > > > > >
> > > > > > >
> > > > > > > Firstly, Method 1 and Method 2 are mathematically equivalent.
> > > > > > >
> > > > > > > Secondly, in the sense of producing computational graph, they are different. In the next, we only consider the terms $\mathbf{W}\_{gnn}^{l} \cdot \mathbf{H}^{l-1}$ and $\mathbf{I} \cdot \mathbf{W}\_{gnn}^{l} \cdot \mathbf{H}^{l-1}$ in Method 1 and 2 since the rest terms are the same. For operation $\mathbf{W}\_{gnn}^{l} \cdot \mathbf{H}^{l-1}$ in Method 1, it only has one dense matrix multiplication. For operation $\mathbf{I} \cdot \mathbf{W}\_{gnn}^{l} \cdot \mathbf{H}^{l-1}$ in Method 2, it has two steps, one is dense matrix multiplication ($\mathbf{Z} = \mathbf{W}\_{gnn}^{l} \cdot \mathbf{H}^{l-1}$), the other is a sparse matrix multiplication ($\mathbf{I} \cdot \mathbf{Z}$). Thus they will produce different computional graph (using torch_geometric). Typically, sparse matrix multiplication needs much more time than dense matrix multiplication (in the case that the sparse matrix is an indentity matrix, this still holds if the computation package has no special optimization for the indentity matrix).
> > > > > > >
> > > > > > > To investigate the efficiency of these two methods in production environments, we conducted experiments to present the running time of different operations ($\mathbf{W} \cdot \mathbf{X}$, $\mathbf{A} \cdot \mathbf{Z}$ and $\mathbf{I} \cdot \mathbf{Z}$) and different methods ($\mathbf{W} \cdot \mathbf{X}$, $\mathbf{A} \cdot \mathbf{W} \cdot \mathbf{X}$ and $\mathbf{I} \cdot \mathbf{W} \cdot \mathbf{X}$). $\mathbf{W} \cdot \mathbf{X}$ is dense matrix multiplication, $\mathbf{A} \cdot \mathbf{Z}$ and $\mathbf{I} \cdot \mathbf{Z}$ are sparse matrix multiplication. The experiments are conducted with an NVIDIA RTX A5000. The softwares and their version are cudatoolkit (11.0.221), PyTorch (1.9.1), torch-sparse (0.6.12).

---

### Official Review · Reviewer_uoW4 · 2022-10-27

**Confidence:** 4
**Clarity, Quality, Novelty And Reproducibility:** The paper is clearly written.
**Correctness:** 3
**Technical Novelty And Significance:** 2
**Empirical Novelty And Significance:** 3
**Recommendation:** 5

**Strength And Weaknesses:**

Strength:
- The proposed method has a strong empirical performance.


Weakness:
- The paper lacks technical novelty. There is no theoretical analysis of the findings.
- The observation is not surprising. All the evaluated datasets have rich feature information, where MLP alone can perform well. Therefore, it's not surprising that a trained MLP weight is a good initialization for GNN. The paper should include more experiments on datasets where node features are less important, e.g., molecule classification.
- More ablation study is needed. To verify the proposed idea is valuable, it will be useful to investigate using shallow-GNN (e.g., 1-layer) weights to initialize GNN.

**Summary Of The Paper:**

This paper proposes a new method for initializing GNN, by utilizing the trained weights from MLP. The method can lead to much faster convergence.

**Summary Of The Review:**

Overall, this paper reveals an interesting observation for GNN initialization. However, I think this is just a good start for a research paper. I highly encourage the authors to derive theoretical analysis based on the finding and include more convincing experiment settings and ablation studies to verify the idea.

---

> ### Author Response · Authors · 2022-11-17
> **Response to Reviewer uoW4 (Part 2/2)**
>
> Moreover, we respectfully argue that the observation found in our paper is surprising and moreover impactful, from the following aspects:
>
> 1. Cross-model weight transferance is intriguing and interesting, yet underexplored. MLPs and GNNs are typically considered as two completely different models in prior literature owing to their clear differences in utilization of neighborhood information, and the observation that weight transferance between them works as well as it does is not obvious, at least to the authors.  This is made further surprising in multi-layer GNN and MLP scenarios, like we consider in most of our experiments (see Table 9). Our work reveals both the feasibility and strong performance of this particular instance of cross-model weight transfer.
> 2. This observation about weight transferance has huge potential to accelerate the training of GNNs on large graphs. Given that MLPs are trained much faster than GNNs, the weight transferance between MLPs and GNNs is of a huge practical impact, and is made even more practical by our simple, accessible and highly effective method, MLPInit.  Our extensive analyses also demonstrate that in addition to speeding up GNN training (Figure 1, Table 3), MLPInit also leads to better quality converged models (Table 4 and 5 for node classification and link prediction, respectively). We also provide two qualitative observations in loss landscape visualization in Section 5.4 (Figure 5), and weight distribution in Appendix A.2, which show MLPInit yields converged models in larger areas of low-loss and with higher-magnitude parameters, promoting generalization.
>
>
> **Q3: The paper should include more experiments on datasets where node features are less important, e.g., molecule classification.**
>
> Thank you for your comment.  Our work focuses on accelerating GNN training, which is mainly useful in the context of training a GNN on a single, large graph with many nodes and edges.  Such tasks are extremely common in large-scale industrial node-level and link-level tasks, including recommendation systems [2,3,4], traffic prediction [5] and forecasting applications [6]. In many of these scenarios, node features are valuable for prediction purposes, but enhanced with graph context. Conversely, molecule classification task is a graph-level task, in which we have multiple small and independent graphs, each with very few nodes. GNN training on molecule classification tasks is typically easier to handle by mini-batch training. Therefore, we consider molecule classification tasks out of the scope and intended focus of our work.
>
>
> **Q4: More ablation study is needed. To verify the proposed idea is valuable, it will be useful to investigate using shallow-GNN (e.g., 1-layer) weights to initialize GNN.**
>
> Thank you for your comment. Could you clarify your question or motivation behind *using shallow-GNN (e.g., 1-layer) weights to initialize GNN*? We do not quite understand the rationale to initialize one GNN with another shallow GNN.  Moreover, unless we consider a special case where all the GNN layers share the same weight matrix, the weight spaces would not be comparable between such settings (the shallow GNN would have fewer weight matrices than the deep one).
>
> [**References**]\
> [1] Dwivedi, Vijay Prakash, Chaitanya K. Joshi, Thomas Laurent, Yoshua Bengio, and Xavier Bresson. "Benchmarking graph neural networks." arXiv preprint arXiv:2003.00982 (2020).\
> [2] He, Xiangnan, Kuan Deng, Xiang Wang, Yan Li, Yongdong Zhang, and Meng Wang. "Lightgcn: Simplifying and powering graph convolution network for recommendation." In Proceedings of the 43rd International ACM SIGIR Conference, pp. 639-648. 2020.\
> [3] Ying, Rex, Ruining He, Kaifeng Chen, Pong Eksombatchai, William L. Hamilton, and Jure Leskovec. "Graph convolutional neural networks for web-scale recommender systems." In Proceedings of the 24th ACM SIGKDD Conference, pp. 974-983. 2018.\
> [4] Sankar, Aravind, Yozen Liu, Jun Yu, and Neil Shah. "Graph neural networks for friend ranking in large-scale social platforms." In Proceedings of the Web Conference 2021, pp. 2535-2546. 2021.\
> [5] Derrow-Pinion, Austin, Jennifer She, David Wong, Oliver Lange, Todd Hester, Luis Perez, Marc Nunkesser et al. "Eta prediction with graph neural networks in google maps." In Proceedings of the 30th ACM CIKM Conference, pp. 3767-3776. 2021\
> [6] Tang, Xianfeng, Yozen Liu, Neil Shah, Xiaolin Shi, Prasenjit Mitra, and Suhang Wang. "Knowing your fate: Friendship, action and temporal explanations for user engagement prediction on social apps." In Proceedings of the 26th ACM SIGKDD Conference, pp. 2269-2279. 2020.
>
> Thanks,\
> Authors

---

> > ### Comment · Reviewer_uoW4 · 2022-11-26
> > **Reply to author response**
> >
> > Thanks for the response.
> >
> > Regarding Q4: My understanding is that an MLP can be viewed as a 0-layer GNN (without message passing). It will be helpful to add an ablation study of initializing a deep GNN with a shallow GNN (e.g., 1-layer), which is a generalized case for a simple MLP. Such a study will be useful to judge how important is feature transformation for the task of interest.

---

> > > ### Author Response · Authors · 2022-11-27
> > > **Response to follow-up questions [Part 2/2]**
> > >
> > >
> > >
> > > **Q2: the new results that the authors report in the rebuttal, where an MLP achieves 53.88 on OGB-arxiv, and GraphSAINT achieves 73.80. However, it is clearly much lower than the results reported in Figure 1 (a random GNN with MLP-Init has a performance of 66 in the figure). Will 1 epoch training provides that much difference?**
> > >
> > >
> > > Thank you for your response and further questions. We want to clarify that the results of GraphSAINT on OGB-arxiv data are not suspicious, but rather **expected and faithful**.
> > >
> > > - Firstly, we note that the number $73.80$ you pointed out is **the result of GraphSAINT on the OGB-products dataset, not OGB-arxiv**. And the accuary of GraphSAINT on the OGB-arxiv dataset is $68.80$.
> > >
> > > - Secondly, the accuracy improvement from $53.88$ (achieved by MLP) to $68.80$ (achieved by MLPInit, not the $73.80$ you mentioned) is not provided by $1$ epoch training. For fine-tuning stage, we initialize GNN with the weights of converged PeerMLP, the accuracy of MLPInit at this moment is $63.26$ (much higher than $53.88$). And after $1$ epoch training, the accuracy of MLPInit is $65.35$ ($66$ you mentioned). Thus, the accuracy improvement by $1$ epoch training is $65.35-63.26=2.09$, which is reasonable and acceptable.
> > >
> > >
> > > To further clarify the impact of training for a single epoch as raised in your question, we also list the accuracies of our method in different training stages of GraphSAINT on OGB-arxiv, including:
> > >
> > > - **MLP**: is PeerMLP to GraphSAINT. The trainable weights of MLP and GraphSAINT are the same.
> > > - **MLPInit at the 0th epoch**: is the GraphSAINT with the weights of the fully-trained PeerMLP, without fine-tuning.
> > > - **MLPInit at the 1st epoch**: is the GraphSAINT with the weights of the fully-trained PeerMLP, with 1-epoch of fine-tuning.
> > > - **MLPInit (final)**: is the best result of our proposed method.
> > >
> > >
> > > Table: Accuracies of MLPInit in different training stages with GraphSAINT on OGB-arxiv dataset.
> > >
> > > | Dataset      | MLP           | MLPInit at the 0th epoch   | MLPInit at the 1st epoch   | MLPInit (final)       |
> > > | --------     | :------------: | :------------: | :------------: |:------------: |
> > > | OGB-arxiv    | 53.88    | 63.26    | 65.35   |  68.80 |
> > >
> > >
> > > From the table, we clearly observe that
> > > - Given the results -- $63.26$ (MLPInit at the 0th epoch) and $65.35$ (MLPInit at the 1st epoch), we observe a $2.09$ difference, which we feel is reasonable and acceptable for single epoch training.
> > >
> > > Additionally,
> > >
> > > - Given that the number $63.26$ (MLPInit at 0th epoch) is larger than $53.88$ (MLP), one can observe that GNN with weights from its converged PeerMLP performs better than the corresponding converged PeerMLP itself (Contrition 1 and Observation 2 in our paper), and supports our claim that *our method can also directly serve as the final, or deployed GNN model, in resource-constrained settings.*
> > > - The difference between the numbers $68.80$ (MLPInit (final)) and $63.26$ (MLPInit at 0th epoch) shows the advantage of fine-tuning, and the value of initializing GNNs using our method.
> > >
> > > We sincerely thank you again for your time and reply. We hope our response has addressed your concerns, you will consider raising your score in light of our responses.
> > >
> > >
> > > Thanks,\
> > > Authors

---

> > > > ### Comment · Reviewer_uoW4 · 2022-11-27
> > > > **Reply to the author response**
> > > >
> > > > > the result of GraphSAINT on the OGB-products dataset, not OGB-arxiv
> > > >
> > > > It was unfortunate that OpenReview does not support uploading figure - but in the previous response, here are the results reported:
> > > >
> > > > > Dataset	Methods	MLP	GNN	Performance gap ratio
> > > > OGB-arxiv	GraphSAGE	56.04	72.00	28.48%
> > > > GraphSAINT	53.88	73.80	36.97%
> > > > ClusterGCN	54.47	78.71	44.50%
> > > > GCN	56.31	77.08	36.89%
> > > >
> > > > The score of 73.80 of GraphSAINT is clearly on OGB-arxiv, right? The results included are misleading at the very least.
> > > >
> > > > > Given that the number 63.26  (MLPInit at 0th epoch) is larger than 53.88 (MLP)
> > > >
> > > > Thanks for spending time to make the clarification. But I have to admit this result confuses me further.
> > > > Here is my rephrasing of the finding: a GNN trained on a graph with no edges (equivalent to an MLP) has a performance of 53.88, after simply applying it to a graph with edges (MLPInit at 0th epoch), the performance significantly boosts to 63.26.
> > > > I hope this rephrase makes sense.
> > > >
> > > > But if this is the case, it suggests that MLPInit is just lucky to get good results, since it has no information on how the graph looks like during training. Let's say we reverse the edges in the graph (make nodes that are connected to disconnect, and nodes that are not connected to connect). In this case MLPInit will generate a very bad initialization.
> > > >
> > > > This thought experiment confirms my previous criticism on the experiments:
> > > > > The paper should include more experiments on datasets where node features are less important
> > > > Meanwhile, I did not find discussions on the limitation of MLPInit in the paper. If the paper had included any theoretical analysis, it will clarify the concerns here.

---

> > > > > ### Author Response · Authors · 2022-11-28
> > > > > **Response to follow-up comments [Part 2/2]**
> > > > >
> > > > >
> > > > >
> > > > >
> > > > > Table: Performance of PeerMLP and GNN with converged  weights of PeerMLP on different $\lambda$, which show different levels of association between node features and labels. The dataset is OGB-arxiv. The accuracy in percentage is based on 5 runs. $\Delta$ is the difference between PeerMLP and GNN with the weights of converged PeerMLP.
> > > > >
> > > > > | $\lambda$                | 0.0     |(0.05) |  0.1  | (0.15)| 0.2     |  0.3  | 0.4     |  0.5 | 0.6     |  0.7 | 0.8     | 0.9  | 1.0    |
> > > > > | --------                 | :----:  |:----: |:----: |:----: | :---:   |:----: |:----:   |:----:|:----:   |:----:|:------: |:----:|:---:   |
> > > > > | PeerMLP                  | 5.95    |6.63   |11.88  |16.39  | 19.82   |23.59  |  25.63  |31.19 | 40.65   |48.74 | 53.40   |55.67 | 56.23  |
> > > > > | GNN with PeerMLP weights | 5.86    |5.86   |5.89   |8.82   | 24.10   |32.87  |  40.67  |51.41 | 55.99   |59.05 | 61.30   |62.14 | 62.43  |
> > > > > | $\Delta$                 | -0.09   |-0.77  |-5.99  |-7.56  | 4.28    |9.29   |  15.05  |20.22 | 15.34   |10.31 | 7.90    |6.47  | 6.20   |
> > > > > | Improv                  | -1.53%  |-11.59%|-50.43%|-46.16%| 21.58%  |39.37% |  58.71% |64.81%| 37.73%  |21.15%| 14.79%  |11.63%| 11.03% |
> > > > >
> > > > >
> > > > >
> > > > > Table: Performance of GNN with Random Init and MLPInit on different $\lambda$. The dataset is OGB-arxiv. The accuracy in percentage is the best performance of the two methods and is based on 5 runs. $Improv$ is the imporvement of GNN with MLPInit to GNN with Random Init.
> > > > >
> > > > > | $\lambda$                | 0.0     |(0.05) |  0.1  | (0.15)| 0.2     |  0.3  | 0.4     |  0.5 | 0.6     |  0.7 | 0.8     | 0.9  | 1.0    | Avg.   |
> > > > > | --------                 | :----:  |:----: |:----: |:----: | :---:   |:----: |:----:   |:----:|:----:   |:----:|:------: |:----:|:---:   |:---:   |
> > > > > | GNN with Random Init     | 60.03   |59.22  |60.71  |63.24  | 64.71   |67.46  | 69.02   | 70.05|70.72    | 71.31| 71.76   | 71.70| 72.00  | 67.07  |
> > > > > | GNN with MLPInit         | 59.78   |60.47  |62.53  |64.49  | 65.94   |67.84  | 69.11   | 70.47|70.70    | 71.45| 71.50   | 71.92| 72.25  | 67.57  |
> > > > > | Improv                   | -0.42%  |2.12%  |3.01%  |1.98%  | 1.91%   |0.57%  | 0.13%   | 0.59%|-0.03%   | 0.20%| -0.36%  | 0.29%| 0.34%  | 0.74%  |
> > > > >
> > > > > Note that we initially conducted the experiments on $\lambda=[0.0,0.1,0.2,...,1.0]$, and we observed that performance on $\lambda=0.1$ is much lower than other values, thus we conducted more on $\lambda=[0.05, 0.15]$ around $0.1$.
> > > > >
> > > > >
> > > > > The results shows that
> > > > >
> > > > > - If node features are uncorrelated to the node labels ($\lambda < 0.2$), GNN with the weights of PeerMLP will not outperform the PeerMLP.
> > > > >
> > > > > - If the node features are correlated to the node labels ($\lambda > 0.2$), GNN with the weights of PeerMLP will consistently outperform the PeerMLP.
> > > > >
> > > > > - Overall, MLPInit obtains a better ($0.74\%$ average improvement) final accuracy than Random Init over $13$ different $\lambda$s.
> > > > >
> > > > >
> > > > > We concluded from above expeirments and the experiments in Tables 3 and 4 as follows:
> > > > >
> > > > > - MLPInit will not provide a good initialization for GNN if the node features give very little information for labels.
> > > > > - Our proposed method performs well in the commonly used standard large datasets, which shows that our proposed method is demonstrably useful in practice.
> > > > >
> > > > >
> > > > >
> > > > > We hope the new experimental results on synthetic graphs will address your concerns about the performance of MLPInt when node features are less important. And we will add the according discussion in the updates version of our paper.
> > > > >
> > > > >
> > > > >
> > > > >
> > > > >
> > > > >
> > > > > **Q5:I did not find discussions on the limitation of MLPInit in the paper. If the paper had included any theoretical analysis, it will clarify the concerns here.**
> > > > >
> > > > > We thank you for your insightful comments, and mentioning these limitations is indeed an oversight.  We will add the limitations to the updated version of our paper.  We observe two limitations:
> > > > >
> > > > > - Our method may not work or be less effective for some graphs, such as (1) reverse the edges (as you mentioned), (2) the label of node is almost irrelevent to the node features (as the results in the new experiments).
> > > > >
> > > > >
> > > > > - Our valuable findings stand to be better supported by theoretical analysis. Theoretical analysis can indeed add value for our work. Although deriving theoretical results about when and how our method doesn't work is likely out-of-scope for this submission given time constraints, we believe there is sufficient room and scope for future work to understand when these types of weight-transferance and initialization strategies work best from a theoretical perspective.
> > > > >
> > > > > We sincerely thank you again for your time and reply. We hope our response and new experiments have addressed some of your concerns.
> > > > >
> > > > >
> > > > > Thanks,\
> > > > > Authors

---

> > > > > > ### Comment · Reviewer_uoW4 · 2022-11-28
> > > > > > **Reply: Thanks for the response**
> > > > > >
> > > > > > I appreciate all the additional author responses; the additional experimental results and discussions are helpful.
> > > > > > Although I still hold my opinion that the paper lacks technical novelty and theoretical results, I can see that the authors truly wish to make the paper better. I would like to increase my score from 3 to 5, and hoping the authors can integrate the discussions into the main content of the paper.

---

> > > > > > > ### Author Response · Authors · 2022-11-29
> > > > > > > **Thank you for raising the score**
> > > > > > >
> > > > > > > Dear Reviewer uow4,
> > > > > > >
> > > > > > > We thank you for your reply and for raising the score. We are glad that our response and new experimental results have addressed some of your concerns. And we will added the new expeirments with the syntheric graphs and the limitations of our method in our revised paper.
> > > > > > >
> > > > > > > We sincerely thank you again for your time and effort in reviewing our paper and replying to our response.
> > > > > > >
> > > > > > > Thanks,\
> > > > > > > Authors

---

> > > > > ### Author Response · Authors · 2022-11-28
> > > > > **Response to follow-up comments [Part 1/2]**
> > > > >
> > > > >
> > > > > Dear Reviewer uoW4,
> > > > >
> > > > > We sincerely thank you for your reply and follow-up questions. We address your concerns as follows:
> > > > >
> > > > > **Q1: in the previous response, here are the results reported**
> > > > >
> > > > > We apologize for the typo in the response text, and we understand the reason for the confusion. The numbers reported in the paper are correct. We update the table as follows, which includes the performance of MLP, GNN with Random Initialization, and the performance gap:
> > > > >
> > > > >
> > > > > | Dataset      | Methods    | MLP       | GNN with Random Init      | Performance gap ratio |
> > > > > | --------     | ---------- | :----:    | :----:  | :------:|
> > > > > | OGB-arxiv    | GraphSAGE  | 56.04     | 72.00   |  28.48% |
> > > > > |              | GraphSAINT | 53.88     | 67.95   |  26.11% |
> > > > > |              | ClusterGCN | 54.47     | 68.00   |  13.53% |
> > > > > |              | GCN        | 56.31     | 70.25   |  24.75% |
> > > > >
> > > > >
> > > > > The results show that GNNs outperform MLPs by a large margin, and a sizeable performance gap between MLPs and GNNs exists.
> > > > >
> > > > > (Note that the $67.95$ is the best accuracy of GraphSAINT with Random Init while the $68.80$ in the previous response is the best accuracy of GraphSAINT with MLPInit)
> > > > >
> > > > >
> > > > >
> > > > >
> > > > > **Q2: my rephrasing of the finding: a GNN trained on a graph with no edges (equivalent to an MLP) has a performance of 53.88, after simply applying it to a graph with edges (MLPInit at 0th epoch), the performance significantly boosts to 63.26. I hope this rephrase makes sense.**
> > > > >
> > > > >
> > > > > Yes, your rephrasing is correct. This is the claimed finding (we think it is quite interesting) in our paper (Contribution 1 and Observation 2): GNN with weights from its converged PeerMLP performs better than the corresponding converged PeerMLP itself (weight transferance between MLPs and GNNs).
> > > > >
> > > > > This finding supports our claims that (1) *Converged weights from PeerMLP provide a good GNN initialization*  (2) *Our method can also directly serve as the final, or deployed GNN model, in resource-constrained settings.*
> > > > >
> > > > >
> > > > >
> > > > > **Q3: But if this is the case, it suggests that MLPInit is just lucky to get good results...Let's say we reverse the edges in the graph (make nodes that are connected to disconnect, and nodes that are not connected to connect). In this case MLPInit will generate a very bad initialization.**
> > > > >
> > > > >
> > > > > Firstly, we completely agree that our proposed method is likely to not work well in some settings, such as (1) reversing the edges (as you mentioned), and (2) the node label is almost irrelevant to the node features.
> > > > >
> > > > > Hwoever, it is also true that our proposed method performs quite well across a variety of  commonly used standard graph ML datasets in node-level and link-level tasks -- on a variety of practical and realistic and commonly-used benchmarks, our method is shown to be demonstrably very useful. Although one could create scenarios where our approach will likely not work as well (e.g., if node features were totally uncorrelated to the labels or if the graph was adversarially constructed), we want to claim that this may be more an exception rather than the rule given the configuration of many practical tasks which form benchmarks in node and link-level tasks in graph ML.  Thus, we would argue that our approach has value in its practicality, despite its limitations in these particular settings which may be less observable or common in practice. Many of these large-scale real-world datasets have properties that are amenable to our method.
> > > > >
> > > > > **Q4: The paper should include more experiments on datasets where node features are less important.**
> > > > >
> > > > >
> > > > > Thanks for this follow-up. Our method (and many others) does depend on a tendency towards node feature-label correlation. Thus, it would likely suffer if features provided less or no information about labels. To further address your concerns in a reasonable way, we conducted experiments on synthetic graphs in lieu of deriving theoretical results given time constraints. Theaw synthetic graphs have differing degrees of correlation between node features and labels.
> > > > >
> > > > > The synthetic graph node features $\mathbf{X}\_{synthetic}$ are generated by mixing the original features and random features for each node in the graph as follows:
> > > > >
> > > > > $\mathbf{X}\_{synthetic}=\lambda\mathbf{X}\_{original}+(1-\lambda)\mathbf{X}\_{random}$
> > > > >
> > > > > where $\mathbf{X}\_{original}$ and $\mathbf{X}\_{random}$ are the original features of OGB-arxiv and random features, and $\lambda$ mediates the two. When $\lambda=0$, the synthetic nodes features will be the original features of OGB-arxiv. When $\lambda=1$, the synthetic nodes features will be random features, which are totally uncorrelated to the node labels. We change the value of $\lambda$ to explore the behavior of MLPInit.

---

> > > ### Author Response · Authors · 2022-11-27
> > > **Response to follow-up questions [Part 1/2]**
> > >
> > >
> > > Dear Reviewer uoW4,
> > >
> > > Thank you very much for your reply and follow-up questions. We address your questions as follows:
> > >
> > >
> > > **Q1: My understanding is that an MLP can be viewed as a 0-layer GNN (without message passing). It will be helpful to add an ablation study of initializing a deep GNN with a shallow GNN (e.g., 1-layer), which is a generalized case for a simple MLP.**
> > >
> > >
> > > Thank you for your clarification for your initial questions. We would like to clarify our proposed method again. MLPInit initializes the GNN with its PeerMLP **layer by layer**. For example, if we take a 4-layer GNN as the target GNN to train, we construct its PeerMLP by making the weights in **each layer** of the GNN and its PeerMLP identical.
> > >
> > > In the following, we present a diagram to show our idea. We use GNNLayer/MLPLayer to denote the layer in GNN/MLP. GNNLayer is $\mathbf{H}^{l} = \sigma( \mathbf{A} \mathbf{W}\_{gnn}^{l} \mathbf{H}^{l-1} )$ as illustrated in our paper (though could be different for different graph convolutions), and MLPLayer is $\mathbf{H}^{l} = \sigma( {\mathbf{W}\_{mlp}^{l}}\mathbf{H}^{l-1})$. Then, the target GNN and its PeerMLP are as follows:
> > >
> > >
> > > The target GNN:
> > > ```
> > > +---------+   +---------+   +---------+   +---------+
> > > |GNNLayer1|-->|GNNLayer2|-->|GNNLayer3|-->|GNNLayer4|
> > > +---------+   +---------+   +---------+   +---------+
> > > ```
> > >
> > > The PeerMLP of the target GNN:
> > > ```
> > > +---------+   +---------+   +---------+   +---------+
> > > |MLPLayer1|-->|MLPLayer2|-->|MLPLayer3|-->|MLPLayer4|
> > > +---------+   +---------+   +---------+   +---------+
> > > ```
> > >
> > >
> > > Since the dimensions of each layer would be different, the trainable weights of GNNLayer are only identical to those of its corresponding MLPLayer, such that GNNLayer1 and MLPLayer1 share the same weights, GNNLayer2 and MLPLayer2 also share the same weights, but are different to those of GNNLayer1. Thus, it is generally infeasible to initialize _all_ the GNNLayers with one GNNLayer (shallow GNN) since the dimensions of the GNNLayers may differ.
> > >
> > >
> > > To further address your concern, we conducted the following experiments on OGB-arxiv dataset by changing the first MLPlayer in PeerMLP to GNNLayer, called PartialPeerMLP.
> > >
> > > The PartialPeerMLP of the target GNN:
> > > ```
> > > +---------+   +---------+   +---------+   +---------+
> > > |GNNLayer1|-->|MLPLayer2|-->|MLPLayer3|-->|MLPLayer4|
> > > +---------+   +---------+   +---------+   +---------+
> > > ```
> > >
> > > We first train PartialPeerMLP to converge and use the converged weights to initialize the target GNN. We present the best accuracy of these two variants of MLPInit and the accuracy of these two MLPInit variants at the 0th training epoch in the following tables:
> > >
> > > Table: Best accuracy of MLPInit variants. The results are based on 10 runs.
> > > | Dataset      | Random Init  | MLPInit with PeerMLP   | MLPInit with PartialPeerMLP   |
> > > | --------     | :------------: | :------------: |:------------: |
> > > | OGB-arxiv    | 72.00         | 72.25         |  72.18       |
> > >
> > >
> > > MLPInit with PartialPeerMLP obtains $72.18$ on OGB-arxiv dataset, which is not better than MLPInit with PeerMLP (even slightly worse). We note that since MLPInit with PartialPeerMLP has one GNNLayer, it involves the graph structure in the pretraining stage, while MLPInit with PeerMLP does not involve graph structure information.
> > >
> > >
> > >
> > > Table: Accuracy of MLPInit variants at the 0th training epoch. The results are based on 10 runs.
> > > | Dataset      | MLPInit with PeerMLP   | MLPInit with PartialPeerMLP   |
> > > | --------     |  :------------: |:------------: |
> > > | OGB-arxiv    |  63.26         |  67.33       |
> > >
> > > The MLPInit with PartialPeerMLP obtains better accuracy than that of  MLPInit with PeerMLP. The results are expected and reasonable since MLPInit with PartialPeerMLP involves the graph structure in the pretraining stage while MLPInit with PeerMLP does not.

---

> > ### Comment · Reviewer_uoW4 · 2022-11-26
> > **Reply**
> >
> > Additionally, I found the additional experimental results provided by the authors suspicious.
> > Concretely, here is the new results that the authors report in the rebuttal, where an MLP achieves 53.88 on OGB-arxiv, and GraphSAINT achieves 73.80. However, it is clearly much lower than the results reported in Figure 1 (a random GNN with MLP-Init has a performance of 66 in the figure). Will 1 epoch training provides that much difference? I hope the authors can better clarify the results

---

> ### Author Response · Authors · 2022-11-17
> **Response to Reviewer uoW4 (Part 1/2)**
>
> Dear Reviewer uoW4,
>
> We thank you for your time and effort in reviewing our paper. Below, we respond to address your concerns.
>
> **Q1:The paper lacks technical novelty. There is no theoretical analysis of the findings.**
>
> Thank you for the comment.  We respectfully argue that our work is novel, and we highlight our technical novelty as follows:
>
> 1. **[Ours is the first work to explore weight transferance between MLPs and GNNs.]** Weight transferance between MLP and GNN has not been explored before. Our work is the first to explore such cross-model weight transferance.
> 2. **[Our work has strong practical implications in GNN training.]** Weight transferance between MLPs and GNNs has solid practical implications. We leverage the observed transferance to accelerate the training of GNNs, and our strong empirical results clearly demonstrate the effectiveness of our proposal, MLPInit.
>
> We understand that added theoretical analysis would make our paper stronger, and agree that our work is largely empirical and practically motivated in nature.  We also understand (by the associated "Summary" section of your review) that in your view, the lack of theoretical analyses strongly contributes towards clear rejection.  However, we believe highly effective and practical empirical contributions which also shed new light on training phenomena are valuable contributions to the field, and should not be conflated with the work's novelty.  The ICLR community also appreciates and encourages these types of works, as per the Review Guidelines (https://iclr.cc/Conferences/2023/ReviewerGuide): *"Submissions bring value to the ICLR community when they convincingly demonstrate new, relevant, impactful knowledge (incl., **empirical**, theoretical, **for practitioners**, etc)."*
>
> We hope that you can consider our work with the merits of what it does aim to contribute to the community: namely, observations around cross-model weight transfer between MLPs and GNNs, and a simple, yet hugely effective and practical initialization method which greatly accelerates GNN training on large-scale graphs, and even improves its performance in many cases.
>
>
> **Q2: The observation is not surprising. All the evaluated datasets have rich feature information, where MLP alone can perform well. Therefore, it's not surprising that a trained MLP weight is a good initialization for GNN.**
>
>
> Thank you for the comment. To the first point, node features are of varying importance in many GNN datasets -- this is made clear by the typical results of GNNs outperforming MLPs (which solely exploit these node features to label correlations) in many papers in the GNN field, including our own (e.g. Table 2).  [1] provides quite comprehensive experimental results comparing GNN to MLP on various datasets and tasks, which substantiates this point.  We also present the performance gap of MLP and GNN in the following table. Note that the MLP and GNN has the same amount of trainable weight in each row of the following table to make a fair comparison. The results show that there is a very large gap in performance of MLPs and GNNs, therefore, GNNs outperform MLPs by a large margin. Still, in almost all of these datasets, sizeable performance gaps between MLPs and GNNs exist, which suggests the claim that "MLP alone can perform well" misses important context.
>
> | Dataset      | Methods    | MLP       | GNN     | Performance gap ratio |
> | --------     | ---------- | :----     | :----   | ------: |
> | OGB-arxiv    | GraphSAGE  | 56.04     | 72.00   |  28.48% |
> |              | GraphSAINT | 53.88     | 73.80   |  36.97% |
> |              | ClusterGCN | 54.47     | 78.71   |  44.50% |
> |              | GCN        | 56.31     | 77.08   |  36.89% |
> | OGB-Products | GraphSAGE  | 63.43     | 80.05   |  26.20% |
> |              | GraphSAINT | 57.29     | 73.80   |  28.82% |
> |              | ClusterGCN | 59.53     | 78.71   |  32.22% |
> |              | GCN        | 62.63     | 77.08   |  23.07% |

---

> ### Author Response · Authors · 2022-11-23
> **Looking forward to your response**
>
> Dear Reviewer uoW4,
>
> Thank you again for the constructive and valuable comments.
>
> As we are nearing the end of the discussion phase, we would like to know if our response has addressed your questions. If you have any further questions, we would be more than happy to address them.
>
> Thank you again for your time. And if our response has addressed your concerns, we hope you will raise your score in light of the revised version and our initial response.
>
> Thanks,\
> Authors

---

### Official Review · Reviewer_gkVw · 2022-11-04

**Confidence:** 5
**Clarity, Quality, Novelty And Reproducibility:** 1. I appreciate the clarity and simpl…
**Correctness:** 3
**Technical Novelty And Significance:** 2
**Empirical Novelty And Significance:** 2
**Recommendation:** 5

**Strength And Weaknesses:**

#### Strengths
1. The idea is easy to describe, understand, and implement. Not only because of these, but the authors also devote much effort to improving the clarity of the writing and the completeness of the experiments.
2. I like the visualizations in Figure 2 (although it is one single case), and the corresponding analysis (i.e., Observation 1). Although not really surprising, these detailed experimental results and visualizations are new to me.
3. I think it is a good idea to consider MLPInit for link-prediction tasks, which may be significantly harder to train due to the stochasticity of the loss. And indeed, it is demonstrated that the performance improvements for link prediction are usually larger.


#### Weaknesses
1. I appreciate the idea that the author wants to make the description of the algorithm simple and leave more space for extensive empirical evaluations. However, the exploration of MLPInit is still a bit limited in the following (but not limited to) aspects:
    1. This paper only considers message-passing GNNs with a fixed and not learnable aggregation operator (or, equivalently, fixed convolution matrix). Some popular MPNNs with learnable, e.g., GAT and GIN-$\epsilon$, or MPNNs with a feature-dependent aggregator, e.g., min/max-aggregations in PNA, are omitted. From the intuition of this paper, it is possible, and likely, MLPInit could also be helpful to speed up the training of those MPNNs with slightly more complexed aggregators, but these discussions or experiments are mostly missing.
    2. Different GNN architectures (e.g., GCN and SAGE) and different graph mini-batch sampling strategies (e.g., GraphSAINT and Cluster-GCN) are actually two separate concepts. I would suggest the authors treat them more separately and discuss them one by one. That is, the generalizability to different GNN architectures and the comparison or combination of different mini-batch sampling strategies should be discussed separately. From the current design of experiment tables, it is unclear what are the two effects independently.
    3. We lack understanding of why MLPInit can improve the final performance of GNNs, even if when GNN is shallow (e.g., 2-layer used in experiments) and the loss/task is simple (e.g., mini-batch trained for node classification). Does it mean without MLPInit, training those GNNs from random initialization will almost always be trapped by some local minima (with higher loss)? Since the authors mostly show the test accuracy but not the training loss, it is a bit unclear whether MLPInit is helpful for convergence or generalization. Also, there are some extreme numbers like node classification on Reddit2 in Table 4, which make people curious to ask if the reported random-initialized GNNs' performance is similar to the other publicly available results. And why could MLPInit simply improve the performance by such a large margin?
    4. The performance improvement for link prediction tasks looks more promising. But again, we lack intuition and understanding of the results. For example, why is the performance in terms of AUC and AP scores similar but in terms of Hits@XX very different?
2. The two observations to motivate MLPInit (section 4 but before section 4.1) seem a bit redundant to me. MLPInit could be a good initialization simply may because the GNNs initialized with MLP weights already have high performance, and the subsequent convergence is much faster than random initialization. I do not quite get the logic/necessity of a separate Observation 2. Why is it necessary that the GNNs with the optimal weights of PeerMLP consistently outperform PeerMLP? Even if it does not outperform PeerMLP, as long as it is better than random initialization, it is already a good initialization to some extent. This redundancy makes this part also a bit hard to understand.
3. The convergence speed comparison (Table 3) compared the number of training epochs (of random initialization and MLPInit) to reach a certain performance. However, it seems in Table 3; we do not consider the number of epochs needed for MLP training, right? So this may make the comparison of Table 3 a bit unfair. Although the epoch-training time of MLP is shorter than the epoch-time of GNN training, since it is not free, we should not simply omit that. To me, the fast convergence is definitely the major claim/contribution of MLPInit, and I suggest the authors be more careful about those experiments.

**Summary Of The Paper:**

This paper leverages the pre-training of a corresponding/analog MLP to speed up the training of GNNs. For most message-passing GNNs, this paper proposes to use the corresponding MLP training as a precursor initialization step to GNN training, which is called the MLPInit. Extensive experiments on multiple large graphs with various GNN architectures validate that MLPInit can accelerate the training of GNNs and also improve the prediction performance for node classification and link prediction tasks.

**Summary Of The Review:**

Overall I would recommend rejection for this current manuscript. This paper explores an interesting direction, initialize some message-passing GNNs' weights using a fully-trained analog MLP, to speed up the convergence and ease the training. However, the discussion is limited to just a few types of fixed-aggregation/convolution GNNs. The convergence speed-up is not reported in a very fair manner. The performance improvement (especially for link prediction) seems promising, but we lack critical understanding, cross-validation, and analysis of the results. Based on these, I think there is still considerable room for improving the manuscript, which could significantly enlarge the potential impact of this work.

---

> ### Author Response · Authors · 2022-11-17
> **Response to Reviewer gkVw (Part 4/4)**
>
> **Q3: The convergence speed comparison (Table 3) compared the number of training epochs (of random initialization and MLPInit) to reach a certain performance. However, it seems in Table 3; we do not consider the number of epochs needed for MLP training, right?**
>
> Thanks for your careful reading. Yes, we do not consider the number of epochs needed for MLP training in Table 3, namely because the training time of MLPs is drastically shorter than that of GNNs (making its contribution negligible to the numbers). We note that running time of MLPs and GNNs largely depends on the concrete implementation. For example, since node features are often a relatively small matrix, and similar for MLP parameters, both can be moved to GPU memory simultaneously and trained in full-batch. To make the comparison more fair and clear, we reported the training epochs needed.
>
>
> To address your concerns, we presented the training time of MLPs/GNNs for one epoch. In this experiment, MLP is trained in a full-batch way and node features are stored in GPU memory. We adopted the official example code of GraphSAGE [9]. The results show that training MLP is much cheaper than training GNN. The running time of MLP only needs $1/147$ and $1/2303$ of GraphSAGE on OGB-arxiv and OGB-Products datasets. In practice, MLP usually only needs to be trained less than $50$ epochs to converge. Thus, the training time of MLP in MLPInit is negligible compared to the training time of GNNs.
>
> Table: Running time comparison of MLP and GraphSAGE (The unit is second. For a fair comparsion, the experiments are conducted on the same computer with NVIDIA RTX A5000.)
> |    Dataset         |    MLP               |    GraphSAGE       |  MLP/GraphSAGE Ratio       |
> | :----------------- | :------------------- | :----------------- | -------------------------: |
> | OGB-arxiv          | 0.03502±0.00004      | 5.170±0.313        | **1/147**                      |
> | OGB-Products       | 0.07631±0.00020      | 175.758±9.560      | **1/2303**                     |
>
>
> Moreover, Table 1 makes a more precise comparison for the computional time of operations in MLPs and GNNs. The results in Table 1 shows that the running time of $Z=WX$ is much smaller than $H=AZ$. MLP only has the operation $Z=WX$  while GNN has two operations, $Z=WX$ and $H=AZ$. $H=AZ$ needs much more computational time (724X, 665X, 3199X) than $Z=WX$, indicating that the runing time of operations in MLP is negligible compared to that of GNNs.
>
>
> **Q4: (Reproducibility): The reproducibility depends on whether the authors will release the code (conditioned on the acceptance) and cannot be judged now.**
>
> Thanks for voicing your concern about our reproducibility. We have provided some simple code for reference in the Supplementary Material along with the submission, and we also presented detailed experimental settings for each experiment  in Appendix C. To further address your concern, we provide the code anonymously at [https://anonymous.4open.science/r/mlpinit-628C](https://anonymous.4open.science/r/mlpinit-628C), and we intend to make it public.
>
> [**References**]\
> [1] Corso, Gabriele, Luca Cavalleri, Dominique Beaini, Pietro Liò, and Petar Veličković. "Principal neighbourhood aggregation for graph nets." Advances in Neural Information Processing Systems 33, 2020.\
> [2] Li, Guohao, Chenxin Xiong, Ali Thabet, and Bernard Ghanem. "Deepergcn: All you need to train deeper gcns." arXiv preprint arXiv:2006.07739 (2020).\
> [3] Li, Hao, Zheng Xu, Gavin Taylor, Christoph Studer, and Tom Goldstein. "Visualizing the loss landscape of neural nets." Advances in neural information processing systems 31, 2018.\
> [4] Duan, Keyu, Zirui Liu, Peihao Wang, Wenqing Zheng, Kaixiong Zhou, Tianlong Chen, Xia Hu, and Zhangyang Wang. "A Comprehensive Study on Large-Scale Graph Training: Benchmarking and Rethinking."  36th NeurIPS Datasets and Benchmarks Track, 2022\
> [5] https://github.com/pyg-team/pytorch_geometric/blob/2.0.4/examples/link_pred.py \
> [6] Zhang, Muhan, and Yixin Chen. "Link prediction based on graph neural networks." Advances in neural information processing systems 31 (2018).\
> [7] Zhang, Muhan, Pan Li, Yinglong Xia, Kai Wang, and Long Jin. "Labeling trick: A theory of using graph neural networks for multi-node representation learning." Advances in Neural Information Processing Systems 34, 2021.\
> [8] Zhao, Tong, Gang Liu, Daheng Wang, Wenhao Yu, and Meng Jiang. "Learning from counterfactual links for link prediction." In International Conference on Machine Learning, 2022.\
> [9] https://github.com/pyg-team/pytorch_geometric/blob/2.0.4/examples/ogbn_products_sage.py
>
>
>
> Thanks,\
> Authors

---

> ### Author Response · Authors · 2022-11-17
> **Response to Reviewer gkVw (Part 3/4)**
>
> **Q1.4: We lack intuition and understanding of the results of link prediction . For example, why is the performance in terms of AUC and AP scores similar but in terms of Hits@XX very different?**
>
> We thank you for your careful reading and observation. The code we used [5] formalizes link prediction as a binary classification task and uses log-loss as the loss function. The metrics we used are as follows:
>
> * AUC and AP are measures for binary classification, reflecting the value of loss for link prediction.
> * Hits@K is the count of how many positive samples are ranked in the top-K positions against a bunch of negative samples.
>
> Previous works [6,7,8] also show that AUC and Hits@k metrics show very different trends and improvement ratios for link prediction because of the above differences.  Tables 1, 2 in [6] show that most link prediction methods seem to do good in AUC, so very high numbers here are common.  Table 1 in [7] and Tables 2, 3, 6, 7 in [8] show that AUC/AP and Hits@k metrics tend to show very different improvement ratios of different methods for link prediction. Typically, AUC/AP has a slight performance improvement ratio, while Hits@k has a larger one. Since both our method and baselines optimize the log loss for link prediction, the improvement of AUC will not be so significant. We conjecture that the node information is beneficial for link prediction, which can be verified to some extent by the results (MLP outperforms GNN on Amazon-products dataset) in Table 6.
>
> In the revised paper, we also added the training curves for link prediction task in Appendix D.3 (Figure 12). The training curves show that MLPInit speeds up the training of link prediction task and can improve Hits@K by a large margin.
>
>
> **Q2: ...Observation 2. Why is it necessary that the GNNs with the optimal weights of PeerMLP consistently outperform PeerMLP? Even if it does not outperform PeerMLP, as long as it is better than random initialization, it is already a good initialization to some extent.**
>
> We agree that it isn't necessary that GNNs with the optimal weights of PeerMLP consistently outperform PeerMLP (both at initialization time, nor after fine-tuning). But, our empirical observations here beget the following advantages:
>
> * We require fewer fine-tuning epochs for the convergence of GNNs, which can reduce the training time of GNNs.
> * It is interesting that GNNs with the optimal weights of PeerMLP consistently outperform PeerMLP, and are able to adopt some of the inductive bias of GNNs without GNN-specific training epochs.
> * It supports our claim that *our method can also directly serve as the final, or deployed GNN model, in resource-constrained settings* (i.e. if we also do not have sufficient time or computational resources to train any GNN epochs). We can only use GNN naively with the optimal weights of PeerMLP (without fine-tuning) if this model consistently outperforms PeerMLP.

---

> ### Author Response · Authors · 2022-11-17
> **Response to Reviewer gkVw (Part 2/4)**
>
> Table: Performance improvements for GNN architectures / layers.
> |  GNN layers| Methods  | Flickr      | Yelp       | Reddit       | Reddit2      | A-products  | OGB-arXiv   | OGB-products | Avg.    |
> | :-------   | :------- | :-----------| :----------| :----------- | :----------- | :---------- | :---------- | :----------- | :----   |
> | SAGEConv   |  Random  | 49.95       | 56.39      | 95.70        | 53.79        | 52.74       | 68.00       | 78.71        | ----    |
> |            |  MLPInit | 49.96       | 58.05      | 96.02        | 77.77        | 55.61       | 69.53       | 78.48        | ↑6.62%  |
> | GCNConv    |  Random  | 50.90       | 40.08      | 92.78        | 27.87        | 36.35       | 70.25       | 77.08        | ----    |
> |            |  MLPInit | 51.16       | 40.83      | 91.40        | 80.37        | 39.70       | 70.35       | 76.85        | ↑14.00%  |
>
>
> From the new form of our results, we observed that 1) MLPInit improves the performance of different graph sampling methods as it improves the performance of GraphSAGE, GraphSAINT and ClusterGCN by $7.96\%$, $7.00\%$ and $6.62\%$. 2) MLPInit improves the performance of different graph neural network layers, as its improve the performance of SAGEConv, GCNConv by $6.62\%$, $14.00\%$.
>
>
>
> **Q1.3: We lack understanding of why MLPInit can improve the final performance of GNNs. Does it mean training those GNNs from random initialization will almost always be trapped by some local minima? Also, there are some extreme numbers like node classification on Reddit2 in Table 4.**
>
> Thanks for the valuable comment. We address your concerns as follows.
>
> **[Understanding of why MLPInit can improve GNNs]** we explain why MLPInit can improve the final GNNs' performance empirically and intuitively.
>
> - Empirically, we provided two analyses to understand why MLPInit can improve the final performance of GNNs. 1) The loss landscape visualization in Section 5.4 (Figure 5) demonstrated that MLPInit results in larger areas of low-loss. 2) The weight distribution in Appendix A.2 shows that with the same number of training epochs, the weights of GraphSAGE with MLPInit are higher-magnitude, promoting generalization [3].
> - Intuitively, node features are task-relevant in many datasets. Training GNNs from scratch with random initialization likely forces the model to learn task-relevant information conditional on neighborhood aggregation. Instead, if a model is trained with MLPInit, the model's training will have two phases: Phase 1 learns solely from the node features, before Phase 2 in which the fine-tuning of the GNN involves neighborhood aggregation.  We suspect that Phase 1 changes the optimization dynamics to find regions of the parameter space guided by only learning with node features, as we observe in Figure 2 and Table 2.
>
> **[GNN trained from random initialization may be trapped by some local minima]** It is difficult to make a claim specifically about random initialization (it may also be true for other initialization techniques, including MLPInit).  That said, the observations (loss landscape visualization in Figure 5, and weight distribution in Appendix A.2) both suggest that MLPInit finds better areas of the parameter space than random initialization. As a corollary, random initialization seems to suffer comparatively due to the smaller low-loss areas around converged parameters, as well as lower-magnitude weights.
>
> **[Result on Reddit2]** Reddit2 is a dataset where node features are so useful, that in fact MLPs can achieve even better performance than GNNs in some cases (See subfigures in the 2nd row, 1st column of Figure 10). From the loss curves in Figure 10, we can see that training GCN from scratch with random initialization struggles to converge. This phenomena demostrates that our method has huge potential for faster convergence for GNNs. It is worth noting that for fair comparison, we adopt the hyperparameters from another paper [4]. And we emphasize that MLPInit is intended to be an acceleration method for GNN training, so the prediction improvement is actually an extra benefit.

---

> ### Author Response · Authors · 2022-11-17
> **Response to Reviewer gkVw (Part 1/4)**
>
> Dear Reviewer gkVw,
>
> We sincerely appreciate your time and constructive and detailed suggestions for our work. We provide the response to your concerns point by point to address your concerns as follows:
>
> **Q1.1: This paper only considers message-passing GNNs with a fixed and not learnable aggregation operator (or, equivalently, fixed convolution matrix). Some popular MPNNs with learnable aggregation operators are omitted.**
>
> Thank you for your suggestion on more complicated and learnable aggregation operators. To address your concern, we conducted experiments with more complicated and learnable aggregation operators. We consider Max [1], Median [1], and Softmax [2] (a learnable aggregation operator). And we presented the performance improvement and speedup in the following tables (Tables 13 and 14 in the updated paper). The dataset used is OGB-arxiv and the GNN backbone is GraphSAGE in this experiment. We use the same pretrained weight of PeerMLP to initialize GraphSAGE with these different aggregation operators.
>
> Table: Performance improvement for GNNs with more complicated and learnable aggregation operators. (The metric is accuracy for node classification.)
> | Methods     | Mean        | Max         | Median      | Softmax     | Avg .   |
> | :-------    | :---------- | :---------- | :---------- | :---------- | :------ |
> | Random      | 72.00±0.16  | 68.31±1.00  | 69.97+0.29  | 71.05±0.20  | 70.33   |
> | MLPInit     | 72.25±0.30  | 69.30+0.56  | 69.95±0.36  | 71.94±0.18  | 70.86   |
> | Improvement | ↑0.36%      | ↑1.44%      | ↓0.02%      | ↑1.25%      | ↑0.75%  |
>
> Table: Speedup for GNNs with more complicated and learnable aggregation operators. (The numbers are the epochs needed by Random Initialization and MLPInit to achieve comparable performance.)
> | Methods     | Mean     | Max    | Median  | Softmax | Avg .  |
> | :-------    | :-----   | :----- | :------ | :-----  | :----- |
> | Random      | 46.7     | 37.1   | 40.9    | 42.0    | 41.6   |
> | MLPInit     | 22.7     | 22.4   | 27.2    | 8.8     | 20.2   |
> | Improvement | 2.06×    | 1.66×  | 1.50×   | 4.77×   | 2.06×  |
>
> The experimental results show that our method can speed up the training of MPNNs with more complicated aggregators and even improve the performance of them. The experimental results also align with your conjecture that *it is possible, and likely, MLPInit could also be helpful to speed up the training of those MPNNs with slightly more complexed aggregators.*
>
> In the revised paper, we added the corresponding discussion and the experiments about the learnable aggregation operators in Appendix D.5. Thanks for suggesting this experiment, as it definitely strengthens our work.
>
> **Q1.2: Different GNN architectures (e.g., GCN and SAGE) and different graph mini-batch sampling strategies (e.g., GraphSAINT and Cluster-GCN) are actually two separate concepts. I would suggest the authors treat them more separately and discuss them one by one.**
>
> We thank you for this valuable suggestion -- we agree that these can be considered separate, though in practice some architectures implicitly assume sampling policies (e.g. GraphSAINT and SAGE).  We provide a form of our result based on your suggestions by separating out results for graph sampling methods, versus specific convolutional layers (e.g., SAGEConv, GCNConv). We also added the new form of results in Appendix D.6.  We retain the consistent improvements in converged model performance, as the below tables show.
>
>
> Table: Performance improvements for graph sampling methods.
> |  Sampling  | Methods  | Flickr      | Yelp       | Reddit       | Reddit2      | A-products  | OGB-arXiv   | OGB-products | Avg.    |
> | :-------   | :------- | :-----------| :----------| :----------- | :----------- | :---------- | :---------- | :----------- | :----   |
> | GraphSAGE  |  Random  | 53.72       | 63.03      | 96.50        | 51.76        | 77.58       | 72.00       | 80.05        | ----    |
> |            |  MLPInit | 53.82       | 63.93      | 96.66        | 89.60        | 77.74       | 72.25       | 80.04        | ↑7.96%  |
> | GraphSAINT |  Random  | 51.37       | 29.42      | 95.58        | 36.45        | 59.31       | 67.95       | 73.80        | ----    |
> |            |  MLPInit | 51.35       | 43.10      | 95.64        | 41.71        | 68.24       | 68.80       | 74.02        | ↑7.00%  |
> | Cluster-GCN|  Random  | 49.95       | 56.39      | 95.70        | 53.79        | 52.74       | 68.00       | 78.71        | ----    |
> |            |  MLPInit | 49.96       | 58.05      | 96.02        | 77.77        | 55.61       | 69.53       | 78.48        | ↑6.62%   |

---

> ### Author Response · Authors · 2022-11-23
> **Looking forward to your response**
>
> Dear Reviewer gkVw,
>
> Thank you again for the constructive and valuable comments.
>
> As we are nearing the end of the discussion phase, we would like to know if our response has addressed your questions. If you have any further questions, we would be more than happy to address them.
>
> Thank you again for your time. And if our response has addressed your concerns, we hope you will raise your score in light of the revised version and our initial response.
>
> Thanks,\
> Authors

---

### Author Response · Authors · 2022-11-17
**Common Response to Reviewers**

Dear Reviewers,

We really appreciate your time and effort in reviewing our work. To address your concerns, we conducted additional experiments and modified our paper accordingly (marked as blue in our revised paper).

* Additional experiments
    * **[Comparison of running time of MLP and GNN, Appendix D.1]** We conducted this experiment to show that training MLP is much cheaper than training GNN. The results show that the running time of MLP can be negligible compared to that of GNNs.
    * **[Loss/accuracy curves of Peermlp and GNN, Appendix D.2]** To investigate the training behavior of PeerMLP and GNN using the weights from trained PeerMLP, we plotted their loss and accuracy curves on training/validation/test sets. The results surprisingly show that GNN using the weight from trained PeerMLP has worse cross-entropy loss but better prediction accuracy than PeerMLP.
    * **[Training curves of link prediction task, Appendix D.3]** To further investigate the training process of link prediction task, we plot the training curves for link prediction task.
    * **[Comparison to GNN pre-training methods, Appendix D.4]** We compare the efficiency of our method to GNN pre-training methods. The comparison to GNN pretraining methods demonstrates the superiority of MLPInit in both effectiveness and efficiency aspects.
    * **[Experiments on more complicated aggregators, Appendix D.5]** We conduct experiments to investigate the effectiveness of MLPInit on GNN with more complicated aggregators.

* Modification of the paper
    * **[Adding performance of MLPInit in Table 2]** The performance gap between GNN and MLPInit clearly show that *MLP-learned parameter can get further improved so we still need to do some training on the GNN*.
    * **[New form of results on graph sampling methods and GNN architectures, Appendix D.6]** We provided the new form of our results in \cref{tab:perf} to show the performance improvements for graph sampling methods and GNN architectures separately.
    * **[Rewriting the Observations 6 and 7]** We modify observations 6 and 7 to make the statements more clear.
    * **[Available code]** The code is anonymously available at [https://anonymous.4open.science/r/mlpinit-628C](https://anonymous.4open.science/r/mlpinit-628C).
    * **[Typos]** We thank all reviewers for pointing out typos and errors; we addressed them carefully.

Thanks again for your efforts in reviewing our work, and we hope our additions (and direct responses) can address any concerns about this work.

Thanks,\
Authors

---

### Decision · Program_Chairs · 2023-01-20

**Decision:**

Accept: poster

**Justification For Why Not Higher Score:**

Some reviewers questioned how generalizable are the observations made in the paper.

**Justification For Why Not Lower Score:**

The reviewers agreed this paper makes an interesting contribution; and one reviewer was particularly enthusiastic about it.

**Metareview: Summary, Strengths And Weaknesses:**

Summary: This paper proposes to accelerate the training of GNNs. This is done by defining an MLP limited to just the node features with a matching parameter count/shapes, training that MLP, and then using the resulting weights to initialize the GNN. Surprisingly, this proves to be very effective, as the MLP weights are close to optimal for the GNN.

The original scores for this paper were 3,3,6,8. After extensive discussion with the authors and between the reviewers, the reviewers updated their score to 5,5,8,8. All reviewers appreciated the simple proposed trick to speed up GNN training. The reviewers who thought the paper was marginally below the acceptance threshold (5) were questioning whether there were sufficient provided evidence for the "generalizability" of MLPInit to other GNN scenarios (see the response to the rebuttal from reviewer gkVw). On the other hand, reviewer pot2 was particularly enthusiastic about the paper in discussion and appreciated the revision with further empirical evaluation. While this AC agrees that further setups could be tested like gkVw suggested, they decided to side with pot2's enthusiasm for the paper as highlighting an interesting line of enquiry to test more and justify this method which already works well. The authors are encouraged to take the discussion in consideration in their camera ready version.

**Note From Pc:**

if the above contains the word "oral" or "spotlight" please see: "oral" presentation means -> notable-top-5% and "spotlight" means -> notable-top-25%. As stated in our emails, we are disassociating presentation type from AC recommendations